# Parallel processing in speech perception with local and global representations of linguistic context

Christian Brodbeck[1,2]*, Shohini Bhattasali[3,4], Aura AL Cruz Heredia[3,5], Philip Resnik[3,4], Jonathan Z Simon[2,6,7], Ellen Lau[3]

[1]Department of Psychological Sciences, University of Connecticut, Storrs, United States; [2]Institute for Systems Research, University of Maryland, College Park, United States; [3]Department of Linguistics, University of Maryland, College Park, United States; [4]Institute for Advanced Computer Studies, University of Maryland, College Park, United States; [5]Department of Psychology, University of Pennsylvania, Philadelphia, United States; [6]Department of Electrical and Computer Engineering, University of Maryland, College Park, United States; [7]Department of Biology, University of Maryland, College Park, United States

**\*For correspondence:**
christianbrodbeck@me.com

**Competing interest:** The authors declare that no competing interests exist.

**Abstract** Speech processing is highly incremental. It is widely accepted that human listeners continuously use the linguistic context to anticipate upcoming concepts, words, and phonemes. However, previous evidence supports two seemingly contradictory models of how a predictive context is integrated with the bottom-up sensory input: Classic psycholinguistic paradigms suggest a two-stage process, in which acoustic input initially leads to local, context-independent representations, which are then quickly integrated with contextual constraints. This contrasts with the view that the brain constructs a single coherent, unified interpretation of the input, which fully integrates available information across representational hierarchies, and thus uses contextual constraints to modulate even the earliest sensory representations. To distinguish these hypotheses, we tested magnetoencephalography responses to continuous narrative speech for signatures of local and unified predictive models. Results provide evidence that listeners employ both types of models in parallel. Two local context models uniquely predict some part of early neural responses, one based on sublexical phoneme sequences, and one based on the phonemes in the current word alone; at the same time, even early responses to phonemes also reflect a unified model that incorporates sentence-level constraints to predict upcoming phonemes. Neural source localization places the anatomical origins of the different predictive models in nonidentical parts of the superior temporal lobes bilaterally, with the right hemisphere showing a relative preference for more local models. These results suggest that speech processing recruits both local and unified predictive models in parallel, reconciling previous disparate findings. Parallel models might make the perceptual system more robust, facilitate processing of unexpected inputs, and serve a function in language acquisition.

## Editor's evaluation

To comprehend speech efficiently, the brain predicts what comes next as sentences unfold. In this study, Brodbeck and colleagues asked at which scale predictive processing helps the analysis of speech. The authors combined magnetoencephalography with state-of-the-art analyses (multivariate Temporal Response Functions) and information-theoretic measures (entropy, surprisal) to test distinct contextual speech models at three hierarchical processing levels. The authors report evidence for

the coexistence of hierarchical and parallel speech processing supporting the independent contribution of local (e.g. sublexical) and global (e.g. sentences) contextual probabilities to the analysis of speech.

## Introduction

Acoustic events in continuous speech occur at a rapid pace, and listeners face pressure to process the speech signal rapidly and incrementally (*Christiansen and Chater, 2016*). One strategy that listeners employ to achieve this is to organize internal representations in such a way as to minimize the processing cost of future language input (*Ferreira and Chantavarin, 2018*). Different accounts have been proposed for how listeners do this, many centered on the notion that they actively predict future input (*Gagnepain et al., 2012*), for instance using internalized generative models (*Halle and Stevens, 1962*). Such predictive strategies manifest in a variety of measures that suggest that more predictable words are easier to process (*Hale, 2003*; *Levy, 2008*; *Smith and Levy, 2013*). For instance, spoken words are recognized more quickly when they are heard in a meaningful context (*Marslen-Wilson and Tyler, 1975*), and words that are made more likely by the context are associated with reduced neural responses, compared to less expected words (*Holcomb and Neville, 2013*; *Connolly and Phillips, 1994*; *Van Petten et al., 1999*; *Diaz and Swaab, 2007*; *Broderick et al., 2018*). This contextual facilitation is pervasive, and is sensitive to language statistics (*Willems et al., 2016*; *Weissbart et al., 2020*; *Schmitt et al., 2020*), as well as the discourse level meaning of the language input for the listeners (*van Berkum et al., 2003*; *Nieuwland and Van Berkum, 2006*).

In speech, words are often predictable because they occur in sequences that form meaningful messages. Similarly, phonemes are predictable because they occur in sequences that form words. For example, after hearing the beginning /ɹɪv/, /ɝ/ would be a likely continuation forming *river*; /i/ would be more surprising, because *riviera* is a less frequent word, whereas /ʊ/ would be highly surprising because there are no common English words starting with that sequence. Phonemes that are thus inconsistent with known word forms elicit an electrophysiological mismatch response (*Gagnepain et al., 2012*), and responses to valid phonemes are proportionally larger the more surprising the phonemes are (*Ettinger et al., 2014*; *Gwilliams and Marantz, 2015*; *Gaston and Marantz, 2017*). Predictive processing is not restricted to linguistic representations, as even responses to acoustic features in early auditory cortex reflect expectations based on the acoustic context (*Singer et al., 2018*; *Forseth et al., 2020*).

Thus, there is little doubt that the brain uses context to facilitate processing of upcoming input, at multiple levels of representation. Here, we investigate a fundamental question about the underlying cognitive organization: Does the brain develop a single, unified representation of the input? In other words, one representation that is consistent across hierarchical levels, effectively propagating information from the sentence context across hierarchical levels to anticipate even low-level features of the sensory input such as phonemes? Or do cognitive subsystems differ in the extent and kind of context they use to interpret their input? This question has appeared in different forms, for example in early debates about whether sensory systems are modular (*Fodor, 1985*), or whether sensory input and contextual constraints are combined immediately in speech perception (*Marslen-Wilson and Tyler, 1975*; *Tanenhaus et al., 1995*). A similar distinction has also surfaced more recently between the local and global architectures of predictive coding (*Tabas and Kriegstein, 2021*).

A strong argument for a unified, globally consistent model comes from Bayesian frameworks, which suggest that, for optimal interpretation of imperfect sensory signals, listeners ought to use the maximum amount of information available to them to compute a prior expectation for upcoming sensory input (*Jurafsky, 1996*; *Norris and McQueen, 2008*). An implication is that speech processing is truly incremental, with a unified linguistic representation that is updated at the phoneme (or an even lower) time scale (*Smith and Levy, 2013*). Such a unified representation is consistent with empirical evidence for top-down modulation of sensory representations, for example, suggesting that recognizing a word can bias subsequent phonetic representations (*Luthra et al., 2021*), that listeners weight cues like a Bayes-optimal observer during speech perception (*Bejjanki et al., 2011*; *Feldman et al., 2009*), and that they immediately interpret incoming speech with regard to communicative goals (*Chambers et al., 2004*; *Heller et al., 2016*). A recent implementation proposed for such a model is the global variant of hierarchical predictive coding, which assumes a cascade of generative

models predicting sensory input from higher-level expectations (*Clark, 2013*; *Friston, 2010*; *Tabas and Kriegstein, 2021*). A unified model is also assumed by classical interactive models of speech processing, which rely on cross-hierarchy interactions to generate a globally consistent interpretation of the input (*McClelland and Rumelhart, 1981*; *McClelland and Elman, 1986*; *Magnuson et al., 2018*).

However, there is also evidence for incomplete use of context in speech perception. Results from cross-modal semantic priming suggest that, during perception of a word, initially multiple meanings are activated regardless of whether they are consistent with the sentence context or not, and contextually appropriate meanings only come to dominate at a later stage (*Swinney, 1979*; *Zwitserlood, 1989*). Similarly, listeners' eye movements suggest that they initially consider word meanings that are impossible given the syntactic context (*Gaston et al., 2020*). Such findings can be interpreted as evidence for a two-stage model of word recognition, in which an earlier retrieval process operates without taking into account the wider sentence context, and only a secondary process of selection determines the best fit with context (*Altmann and Steedman, 1988*). Similarly, at the sublexical level, experiments with nonwords suggest that phoneme sequence probabilities can have effects that are decoupled from the word recognition process (*Vitevitch and Luce, 1999*; *Vitevitch and Luce, 2016*). However, it is also possible that such effects occur only due to the unnaturalness of experimental tasks. For example, in the cross-modal priming task, listeners might come to expect a visual target which is not subject to sentence context constraints, and thus change their reliance on that context.

Finally, a third possibility is that a unified model coexists with more local models of context, and that they operate in a parallel fashion. For example, it has been suggested that the two hemispheres differ with respect to their use of context, with the left hemisphere relying heavily on top-down predictions, and the right hemisphere processing language in a more bottom-up manner (*Federmeier, 2007*).

Distinguishing among these possibilities requires a task that encourages naturalistic engagement with the context, and a nonintrusive measure of linguistic processing. To achieve this, we analyzed magnetoencephalography (MEG) responses to continuous narrative speech, using methods that have previously shown electrophysiological brain responses related to predictive language models. Previous work, however, has tested either only for a local, or only for a unified context model, by either using only the current word up to the current phoneme as context (*Brodbeck et al., 2018a*; *Gillis et al., 2021*) or by using predictions from a complete history of phonemes and words (*Donhauser and Baillet, 2020*). Because these two context models include overlapping sets of constraints, their predictions for neural responses are correlated, and thus need to be assessed jointly. Furthermore, some architectures predict that both kinds of context model should affect brain responses separately. For example, a two-stage architecture predicts an earlier stage of lexical processing that is sensitive to lexical statistics only, and a later stage that is sensitive to the global sentence context. Here, we directly test such possibilities by comparing the ability of different context models to jointly predict brain responses.

## Expressing the use of context through information theory

The sensitivity of speech processing to different definitions of context is formalized through conditional probability distributions (*Figure 1*). Each distribution reflects an interpretation of ongoing speech input, at a given level of representation. We here use word forms and phonemes as units of representation (*Figure 1A*), and all our predictors reflect information-theoretic quantities at the rate of phonemes; however, this is a matter of methodological convenience, and similar models could be formulated using a different granularity (*Smith and Levy, 2013*). *Figure 1B* shows an architecture in which each level uses local information from that level, but information from higher levels does not affect beliefs at lower levels. In this architecture, phonemes are classified at the sublexical level based on the acoustic input and possibly a local phoneme history. The word level decodes the current word from the incoming phonemes, but without access to the multiword context. Finally, the sentence-level updates the sentence representation from the incoming word candidates, and thus selects those candidates that are consistent with the sentence context. In such a model, apparent top-down effects such as perceptual restoration of noisy input (*Ganong, 1980*; *Leonard et al., 2016*) are generated at higher-level decision stages rather than at the initial perceptual representations (*Norris, 1994*). In contrast, *Figure 1C* illustrates the hypothesis of a unified, global context model, in which priors at lower levels take advantage of information available at the higher levels. Here, the sentence context

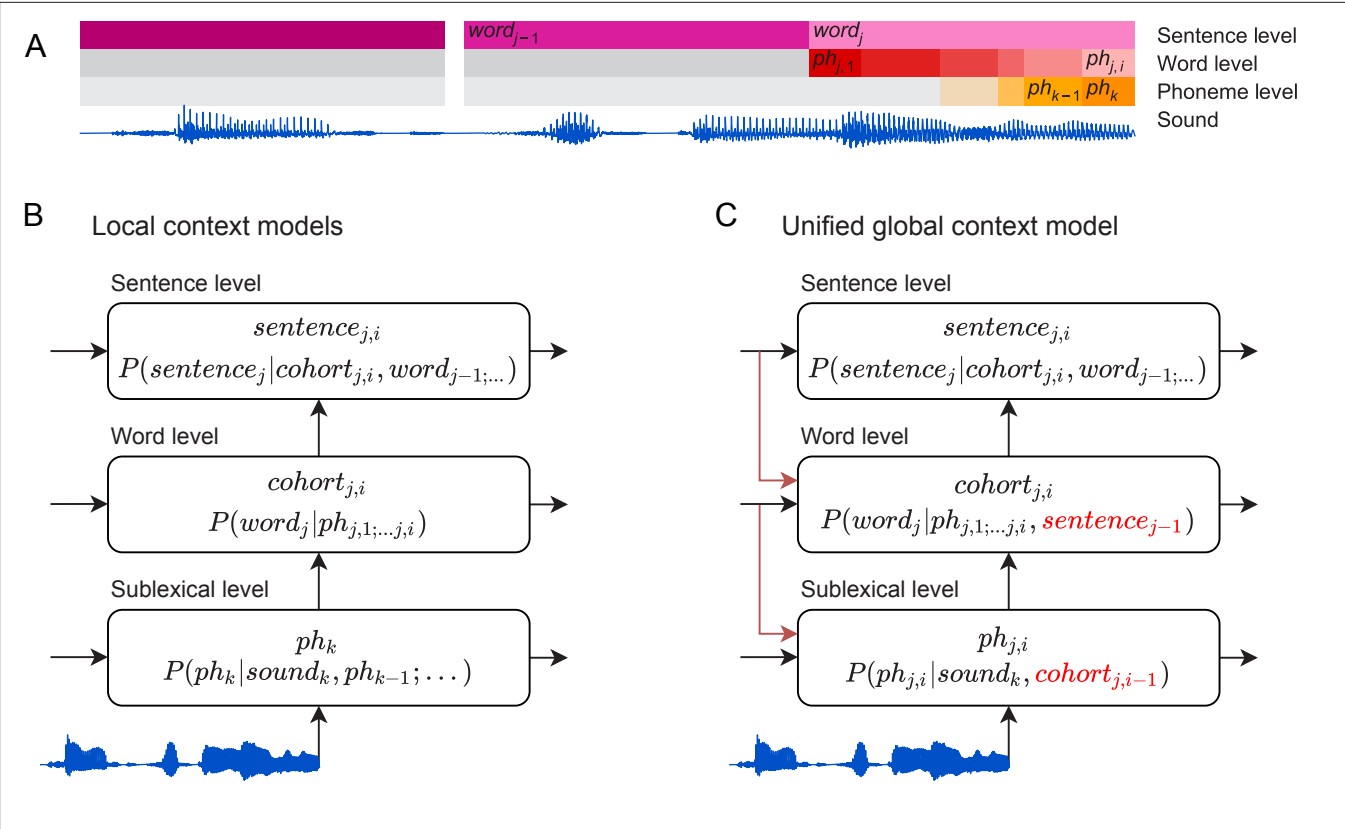

**Figure 1.** Information flow in local and unified architectures for speech processing. (**A**) Schematic characterization of the linguistic units used to characterize speech. The same phoneme can be invoked as part of a sublexical phoneme sequence, $ph_k$, or as part of $word_j$, $ph_{j,i}$. (**B**) Each box stands for a level of representation, characterized by its output and a probability distribution describing the level's use of context. For example, the sublexical level's output is an estimate of the current phoneme, $ph_k$, and the distribution for $ph_k$ is estimated as probability for different phonemes based on the sound input and a sublexical phoneme history. At the sentence level, $sentence_{j,i}$ stands for a temporary representation of the sentence at time $j,i$. Boxes represent functional organization rather than specific brain regions. Arrows reflect the flow of information: each level of representation is updated incrementally, combining information from the same level at the previous time step (horizontal arrows) and the level below (bottom-up arrows). (**C**) The unified architecture implements a unified, global context model through information flowing down the hierarchy, such that expectations at lower levels incorporate information accumulated at the sentence level. Relevant differences from the local context model are in red. Note that while the arrows only cross one level at a time, the information is propagated in steps and eventually crosses all levels.

is used in decoding the current word by directly altering the prior expectation of the word candidates, and this sentence-appropriate expectation is in turn used to alter expectations for upcoming phonemes.

These hypotheses make different predictions for brain responses sensitive to language statistics. Probabilistic speech representations, as in *Figure 1*, are linked to brain activity through information-theoretic complexity metrics (*Hale, 2016*). The most common linking variable is *surprisal*, which is equivalent to the difficulty incurred in updating an incremental representation of the input (*Levy, 2008*). Formally, the surprisal experienced at phoneme *k* is inversely related to the likelihood of that phoneme in its context:

$$I\left(ph_k\right) = -log_2\left(p\left(ph_k|context\right)\right)$$

A second information-theoretic measure that has been found to independently predict brain activity is entropy (*Brodbeck et al., 2018a*; *Donhauser and Baillet, 2020*), a measure of the uncertainty in a probability distribution. Phoneme entropy is defined as the expected (average) surprisal for the next phoneme:

$$H_{ph}\left(ph_k\right) = -\sum_{ph}^{phonemes} p\left(ph_{k+1} = ph|context\right) \log_2\left(p\left(ph_{k+1} = ph|context\right)\right)$$

In contrast to surprisal, which is a function of the expected probability of the current event only, entropy is a function of the whole distribution of expectations. This makes it possible to distinguish between phoneme entropy, the uncertainty about the next phoneme, and cohort entropy, the uncertainty about the complete word form that matches the current partial input (for more details see *Lexical context model* in *Methods*):

$$H_w\left(ph_{j,i}\right) = -\sum_{word}^{lexicon} p\left(word_j = word|context\right) \log_2\left(p\left(word_j = word|context\right)\right)$$

Entropy might relate to neuronal processes in at least two ways. First, the amount of uncertainty might reflect the amount of competition among different representations, which might play out through a neural process such as lateral inhibition (*McClelland and Elman, 1986*). Second, uncertainty might also be associated with increased sensitivity to bottom-up input, because the input is expected to be more informative (*Jaramillo and Zador, 2011*; *Auksztulewicz et al., 2019*).

## Models for responses to continuous speech

To test how context is used in continuous speech processing, we compared the ability of three different context models to predict MEG responses, corresponding to the three levels in *Figure 1B* (see *Figure 2*). The context models all incrementally estimate a probability distribution at each phoneme position, but they differ in the amount and kind of context they incorporate. Throughout, we used *n*-gram models to estimate sequential dependencies because they are powerful language models that can capture effects of language statistics in a transparent manner, with minimal assumptions about the underlying cognitive architecture (*Futrell et al., 2020*; *Levy, 2008*; *Smith and Levy, 2013*). An example of the complete set of predictors is shown in *Figure 3*.

### Sublexical context model

A 5 gram model estimates the prior probability for the next phoneme given the four preceding phonemes. This model reflects simple phoneme sequence statistics (*Vitevitch and Luce, 1999*; *Vitevitch and Luce, 2016*) and is unaware of word boundaries. Such a model is thought to play an important role in language acquisition (*Cairns et al., 1997*; *Chambers et al., 2003*; *Saffran et al., 1996*), but it is unknown whether it has a functional role in adult speech processing. The sublexical model predicted brain responses via the phoneme surprisal and entropy linking variables.

### Word context model

This model implements the cohort model of word perception (*Marslen-Wilson, 1987*), applied to each word in isolation. The first phoneme of the word generates a probability distribution over the lexicon, including all words starting with the given phoneme, and each word's probability proportional to the word's relative unigram frequency. Each subsequent phoneme trims this distribution by removing words that are inconsistent with that phoneme. Like the sublexical model, the lexical model can be used as a predictive model for upcoming phonemes, yielding phoneme surprisal and entropy variables. In addition, the lexical model generates a probability distribution over the lexicon, which yields a cohort entropy variable.

### Sentence context model

The sentence model is closely related to the word context model, but each word's prior probability is estimated from a lexical 5 gram model. While a 5 gram model misses longer-range linguistic dependencies, we use it here as a conservative approximation of sentence-level linguistic and interpretive constraints (*Smith and Levy, 2013*). The sentence model implements cross-hierarchy predictions by using the sentence context in concert with the partial current word to predict upcoming phonemes. Brain activity is predicted from the same three variables as from the word context model.

We evaluated these different context models in terms of their ability to explain held-out MEG responses, and the latency of the brain responses associated with each model. An architecture based on local context models, as in *Figure 1B*, predicts a temporal sequence of responses as information passes up the hierarchy, with earlier responses reflecting lower order context models. In contrast, a unified architecture, as in *Figure 1C*, predicts that the sentence context model should exhaustively explain brain responses, because all representational levels use priors derived from the sentence

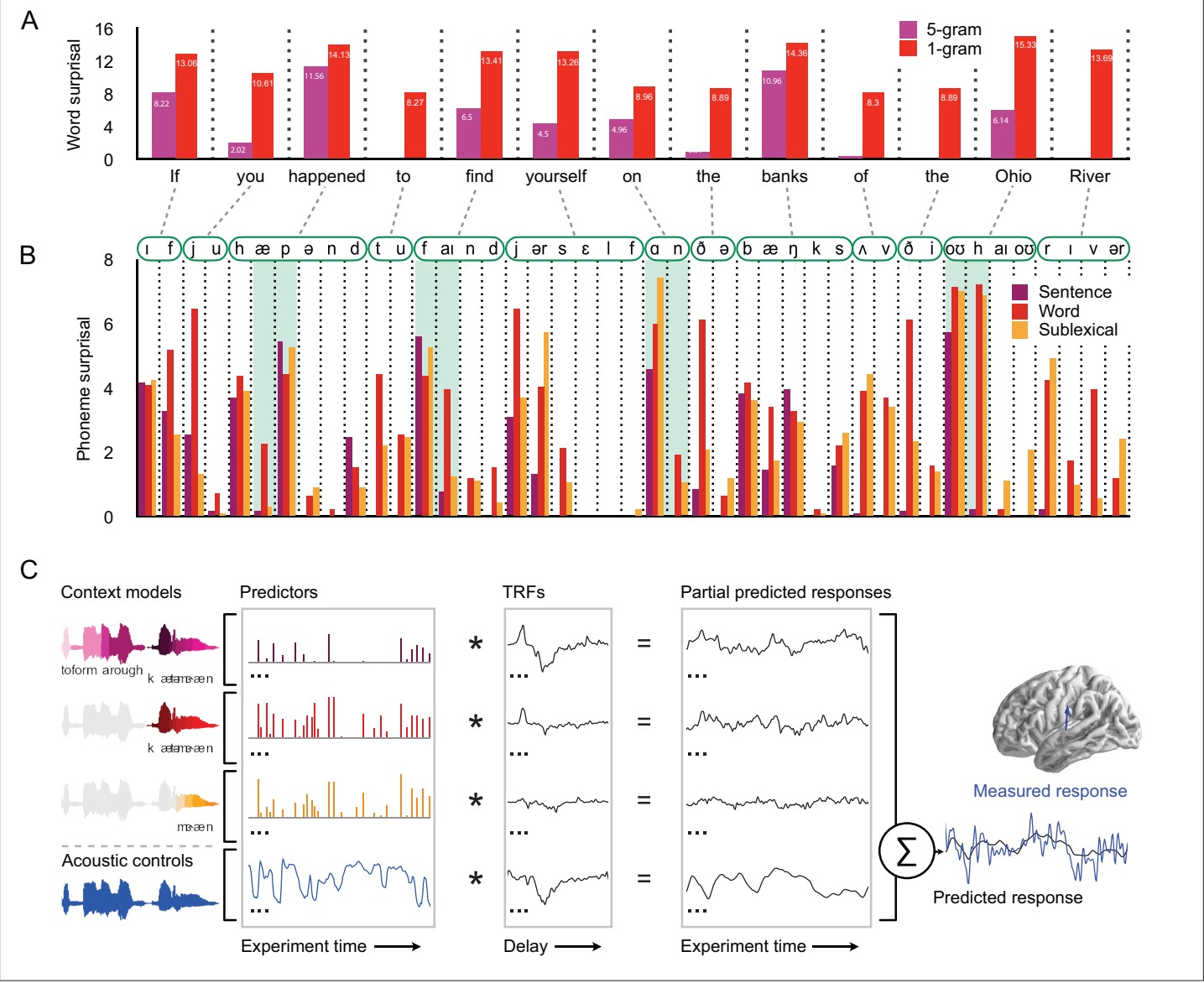

**Figure 2.** Models for predictive speech processing based on the sentence, word, and sublexical context, used to predict magnetoencephalography (MEG) data. (**A**) Example of word-by-word surprisal. The sentence (5 gram) context generally leads to a reduction of word surprisal, but the magnitude of the reduction differs substantially between words (across all stimuli, mean ± standard deviation, unigram surprisal: 10.76 ± 5.15; 5 gram surprisal: 7.43 ± 5.98; $t_{8172}$ = 76.63, p < 0.001). (**B**) Sentence-level predictions propagate to phoneme surprisal, but not in a linear fashion. For example, in the word *happened*, the phoneme surprisal based on all three models is relatively low for the second phoneme /æ/ due to the high likelihood of word candidates like *have* and *had*. However, the next phoneme is /p/ and phoneme surprisal is high across all three models. On the other hand, for words like *find*, *on*, and *Ohio*, the sentence-constrained phoneme surprisal is disproportionately low for subsequent phonemes, reflecting successful combination of the sentence constraint with the first phoneme. (**C**) Phoneme-by-phoneme estimates of information processing demands, based on different context models, were used to predict MEG responses through multivariate temporal response functions (mTRFs) (*Brodbeck et al., 2018b*). An mTRF consists of multiple TRFs estimated jointly such that each predictor, convolved with the corresponding TRF, predicts a partial response, and the pointwise sum of partial responses constitutes the predicted MEG response. The dependent measure (measured response) was the fixed orientation, distributed minimum norm source current estimate of the continuous MEG response. The blue arrow illustrates a single virtual current source dipole. Estimated signals at current dipoles across the brain were analyzed using a mass-univariate approach. See Methods for details. TRFs: temporal response functions.

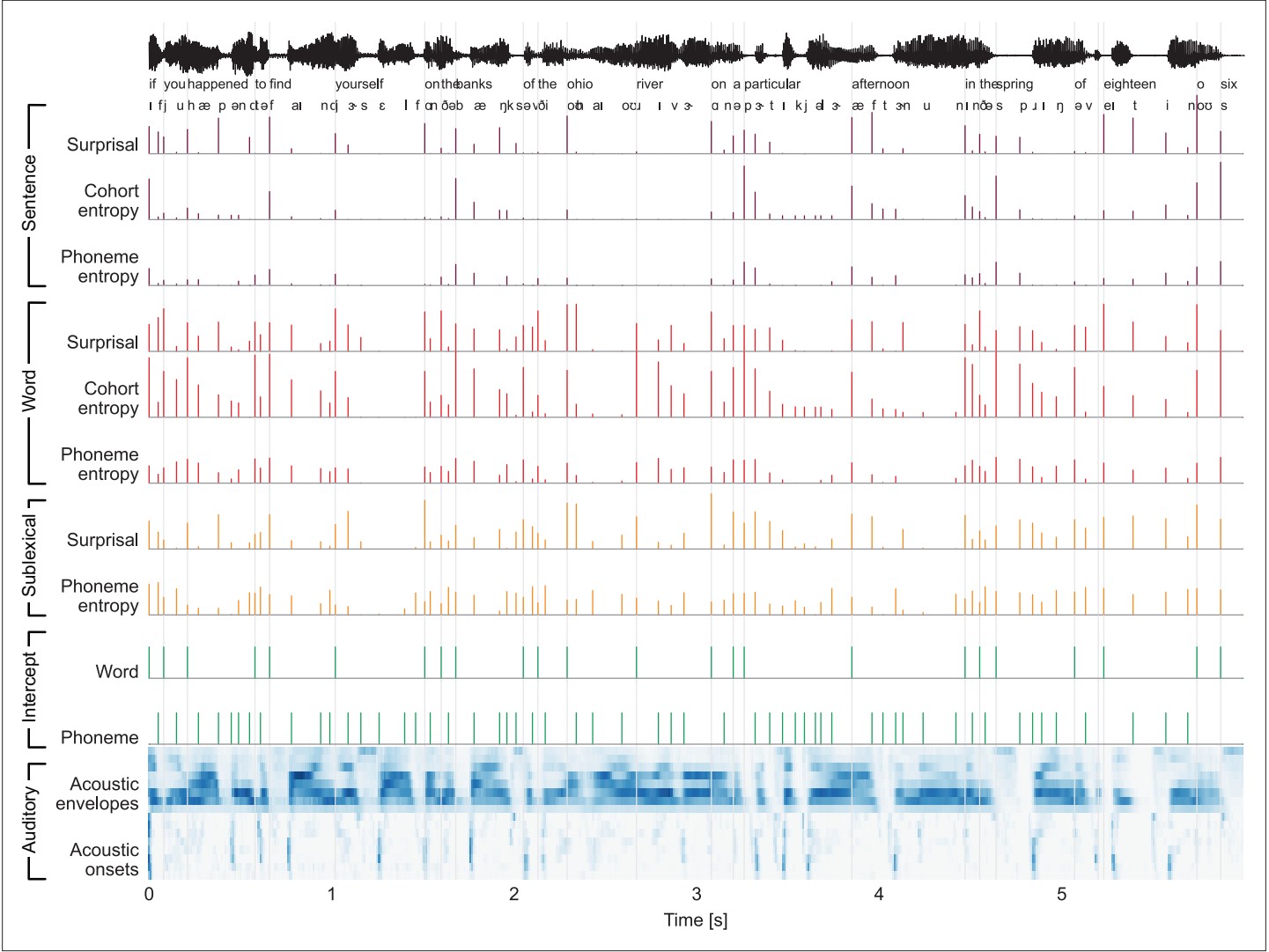

**Figure 3.** Stimulus excerpt with all 26 predictors used to model brain responses. From the top: phoneme-level information-theoretic variables, based on different context definitions: sentence, word, and sublexical context; intercepts for word- and phoneme-related brain activity, that is, a predictor to control for brain activity that does not scale with the variables under consideration; and an auditory processing model, consisting of an acoustic spectrogram (sound envelopes) and an onset spectrogram (sound onsets), each represented by eight predictors for eight frequency bands.

context. Finally, architectures that entail multiple kinds of models predict that different context models might explain response components, possibly in different anatomical areas.

### Acoustic controls
In order to dissociate effects of linguistic processing from responses to acoustic properties of the speech stimulus, all models controlled for a gammatone spectrogram and an acoustic onset spectrogram (***Brodbeck et al., 2020***), as well as word onsets and phoneme onsets (***Brodbeck et al., 2018a***).

## Results
Twelve participants listened to ~47 min of a nonfiction audiobook. Multivariate temporal response functions (mTRFs) were used to jointly predict held-out, source localized MEG responses (***Figure 2C***). To test whether each context model is represented neurally, the predictive power of the full model including all predictors was compared with the predictive power of a model that was estimated without the predictor variables belonging to this specific context model. Besides 11 right-handers, our sample included a single left-hander. While this participant's brain responses were more right-lateralized than

average, excluding them did not change the conclusions from any of the reported lateralization significance tests. We thus report results from the total sample, but identify the left-hander in plots and source data.

## Phoneme-, word-, and sentence-constrained models coexist in the brain

Each context model significantly improves the prediction of held-out data, even after controlling for acoustic features and the other two context models (*Figure 4A*). Each of the three context models' source localization is consistent with sources in the superior temporal gyrus (STG), thought to support phonetic and phonological processing (*Mesgarani et al., 2014*). In addition, the sentence-constrained model also extends to more ventral parts of the temporal lobe, consistent with higher-level language processing (*Hickok and Poeppel, 2007*; *Wilson et al., 2018*). In comparison, the predictive power of the acoustic features is highest in closer vicinity of Heschl's gyrus (*Figure 4D*). At each level of context, surprisal and entropy contribute about equally to the model's predictive power (*Figure 4B*, *Table 1*).

Overall, the acoustic features explain more of the variability in brain responses than the linguistic features (compare scales in *Figure 4A–D*). This is likely because speech is an acoustically rich stimulus, driving many kinds of auditory receptive fields. In contrast, the linguistic predictors represent very specific computations, likely represented in a small and specialized neural territory. For the present purpose, what is critical is the consistency of the linguistic effects across subjects: *Figure 4B,C*, as well as the effect sized shown in *Table 1* suggests that the linguistic modulation of brain responses can be detected very reliably across subjects.

The significant predictive power of the local context models is inconsistent with the hypothesis of a single, unified context model (*Figure 1C*). Instead, it suggests that different neural representations incorporate different kinds of context. We next pursued the question of how these different representations are organized hierarchically. Phoneme surprisal depends on the conditional probability of the current phoneme, and thus does not distinguish between whether what is predicted is a single phoneme or the whole lexical completion (*Levy, 2008*; *Smith and Levy, 2013*). Entropy, on the other hand, depends on the units over which probabilities are calculated, and can thus potentially distinguish between whether brain responses reflect uncertainty over the next phoneme alone, or uncertainty over the word currently being perceived, that is, over the lexical completion (see *Lexical context model* in *Methods*). This distinction is particularly interesting for the sentence context model: if predictions are constrained to using context *within* a hierarchical level, as in *Figure 1B*, then the sentence context should affect uncertainty about the upcoming word, but not uncertainty about the upcoming phoneme. On the other hand, a brain response related to sentence-conditional phoneme entropy would constitute evidence for cross-hierarchy predictions, with sentence-level context informing predictions of upcoming phonemes.

Even though phoneme and cohort entropy were highly correlated (sentence context: $r = 0.92$; word context: $r = 0.90$), each of the four representations was able to explain unique variability in the MEG responses that could not be attributed to any of the other representations (*Figure 4C*, *Table 1*). This suggests that the sentence context model is not restricted to predicting upcoming words, but also generates expectations for upcoming phonemes. This is thus evidence for cross-hierarchy top-down information flow, indicative of a unified language model that aligns representations across hierarchical levels. Together, these results thus indicate that the brain does maintain a unified context model, but that it *also* maintains more local context models.

## Different context models affect different neural processes

All three context models individually contribute to neural representations, but are these representations functionally separable? While all three context models improve predictions in both hemispheres, the sentence-constrained model does so symmetrically, whereas the lexical and sublexical models are both more strongly represented in the right hemisphere than in the left hemisphere (*Figure 4A*). This pattern might suggest an overall right lateralization of linguistic processing; however, the predictive power of the joint linguistic model (predictors from all three levels combined) is not significantly lateralized ($t_{max} = 4.11$, $p = 0.134$). These results thus suggest that linguistic processing is bilateral, but that the hemispheres differ in what context models they rely on. Consistent with this, the context models differ in their relative lateralization (*Figure 4E*). The sublexical context model is significantly more right-lateralized than the sentence model ($t_{11} = 4.41$, $p = 0.001$), while the word model is only

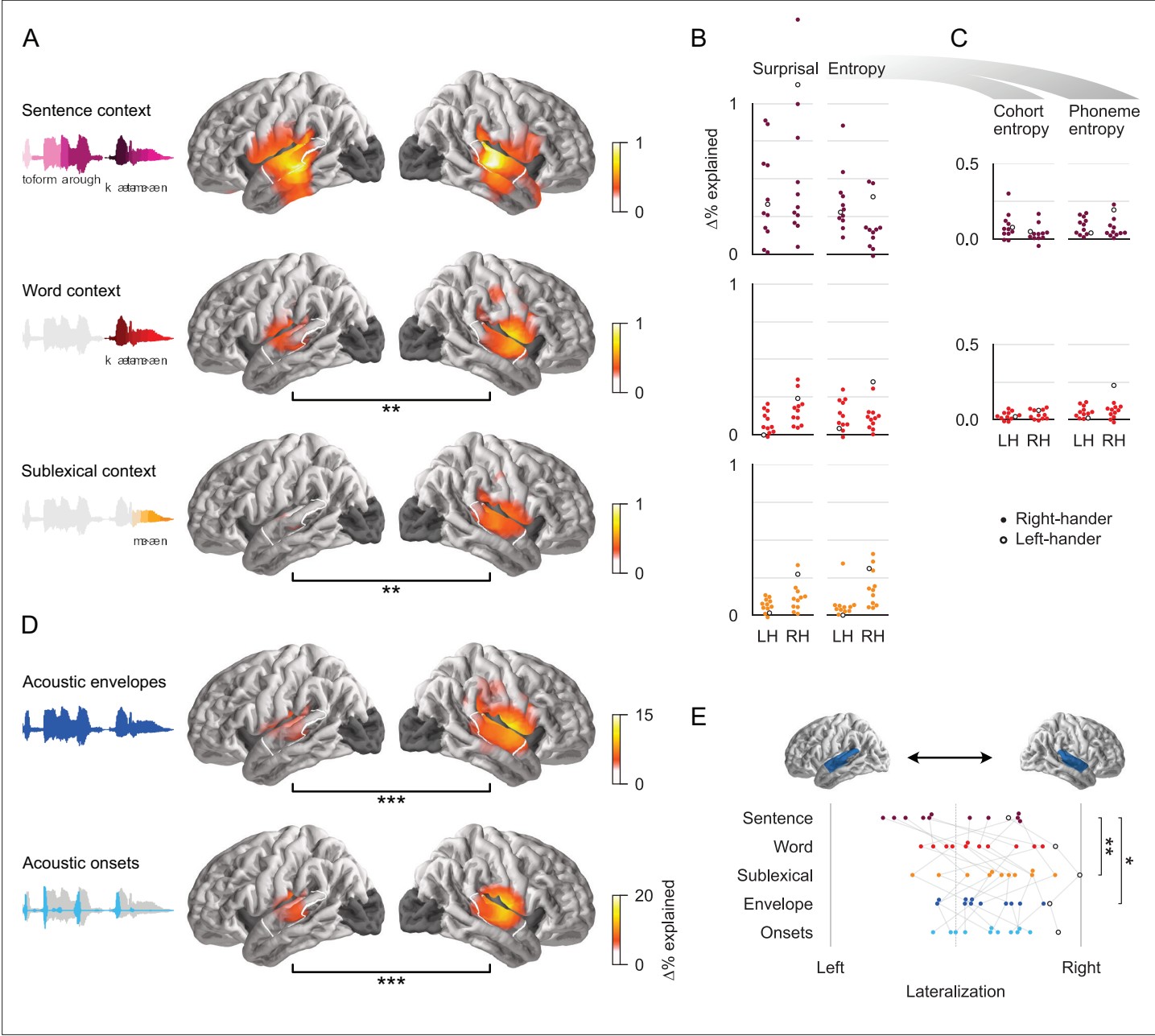

**Figure 4.** All context models significantly contribute to predictions of brain responses. (**A**) Each context model significantly improves predictions of held-out magnetoencephalography (MEG) data in both hemispheres ($t_{max} \geq 6.16$, $p \leq 0.005$). Black bars below anatomical plots indicate a significant difference between hemispheres. The white outline indicates a region of interest (ROI) used for measures shown in (**B**), (**C**), and (**E**). Brain regions excluded from analysis are darkened (occipital lobe and insula). (**B**) Surprisal and entropy have similar predictive power in each context model. Each dot represents the difference in predictive power between the full and a reduced model for one subject, averaged in the ROI. Cohort- and phoneme entropy are combined here because the predictors are highly correlated and hence share a large portion of their explanatory power. Corresponding statistics and effect size are given in *Table 1*. A single left-handed participant is highlighted throughout with an unfilled circle. LH: left hemisphere; RH: right hemisphere. (**C**) Even when tested individually, excluding variability that is shared between the two, cohort- and phoneme entropy at each level significantly improve predictions. A significant effect of sentence-constrained phoneme entropy is evidence for cross-hierarchy integration, as it suggests that sentence-level information is used to predict upcoming phonemes. (**D**) Predictive power of the acoustic feature representations. (**E**) The lateralization index ($LI = R / (L + R)$) indicates that the sublexical context model is more right-lateralized than the sentence context model. Left: LI = 0; right: LI = 1. Significance levels: *$p \leq 0.05$; **$p \leq 0.01$; ***$p \leq 0.001$.

The online version of this article includes the following source data for figure 4:

**Source data 1.** Mass-univariate statistics results for Panels A & D.

**Source data 2.** Predictive power in the mid/posterior superior temporal gyrus ROI, data used in Panels B, C & E.

**Table 1.** Predictive power in mid/posterior superior temporal gyrus region of interest for individual predictors.

One-tailed *t*-tests and Cohen's *d* for the predictive power uniquely attributable to the respective predictors. Data underlying these values are the same as for the swarm plots in *Figure 4B, C* (*Figure 4—source data 2*). *p ≤ 0.05; **p ≤ 0.01; ***p ≤ 0.001.

| | Left hemisphere | | | | Right hemisphere | | | |
|---|---|---|---|---|---|---|---|---|
| | Δ‰ | *t*(11) | p | *d* | Δ‰ | *t*(11) | p | *d* |
| **Sentence context** | | | | | | | | |
| Surprisal | 3.77 | 4.40** | 0.001 | 1.27 | 5.51 | 4.14** | 0.002 | 1.19 |
| Entropy | 3.40 | 5.96*** | <0.001 | 1.72 | 1.94 | 4.11** | 0.002 | 1.19 |
| Cohort | 0.83 | 3.41** | 0.006 | 0.98 | 0.39 | 2.45* | 0.032 | 0.71 |
| Phoneme | 0.85 | 5.18*** | <0.001 | 1.50 | 0.79 | 3.85** | 0.003 | 1.11 |
| **Word context** | | | | | | | | |
| Surprisal | 0.78 | 3.62** | 0.004 | 1.04 | 1.71 | 5.76*** | <0.001 | 1.66 |
| Entropy | 1.26 | 4.43** | 0.001 | 1.28 | 1.31 | 4.39** | 0.001 | 1.27 |
| Cohort | 0.25 | 3.29** | 0.007 | 0.95 | 0.36 | 3.99** | 0.002 | 1.15 |
| Phoneme | 0.51 | 4.59*** | <0.001 | 1.32 | 0.66 | 3.61** | 0.004 | 1.04 |
| **Sublexical context** | | | | | | | | |
| Surprisal | 0.64 | 4.88*** | <0.001 | 1.41 | 1.29 | 4.59*** | <0.001 | 1.33 |
| Entropy | 0.66 | 2.53* | 0.028 | 0.73 | 1.91 | 5.32*** | <0.001 | 1.54 |

numerically more right-lateralized than the sentence model ($t_{11}$ = 1.53, p = 0.154). These lateralization patterns suggest an anatomical differentiation in the representations of different context models, with the left hemisphere primarily relying on a unified model of the sentence context, and the right hemisphere more broadly keeping track of different context levels.

Given that all three context models are represented in the STG, especially in the right hemisphere, a separate question concerns whether, within a hemisphere, the different context models predict activity in the same or different neural sources. While MEG source localization does not allow precisely separating different sources in close proximity, it does allow statistically testing whether two effects originate from the same or from a different configuration of neural sources (*Lütkenhöner, 2003*). The null hypothesis of such a test (*McCarthy and Wood, 1985*) is that a single neural process, corresponding to a fixed configuration of current sources, generates activity that is correlated with all three context models. The alternative hypothesis suggests some differentiation between the configuration of sources recruited by the different models. Results indicate that, in the right hemisphere, all three context models, as well as the two acoustic models, originate from different source configurations ($F_{(175, 1925)}$ ≥ 1.25, p ≤ 0.017). In the left hemisphere, the sentence-constrained model is localized differently from all other models ($F_{(179, 1969)}$ ≥ 1.38, p < 0.001), whereas there is no significant distinction among the other models (possibly due to lower power due to the weaker effects in the left hemisphere for all but the sentence model). In sum, these results suggest that the different context models are maintained by at least partially separable neural processes.

## Sentence context affects early responses and dominates late responses

The temporal response functions (TRFs) estimated for the full model quantify the influence of each predictor variable on brain responses over a range of latencies (*Figure 2C*). *Figure 5* shows the response magnitude to each predictor variable as a function of time, relative to phoneme onset. For an even comparison between predictors, TRFs were summed in the anatomical region in which any context model significantly improved predictions. Note that responses prior to 0 ms are plausible due to coarticulation, by which information about a phoneme's identity can already be present in the

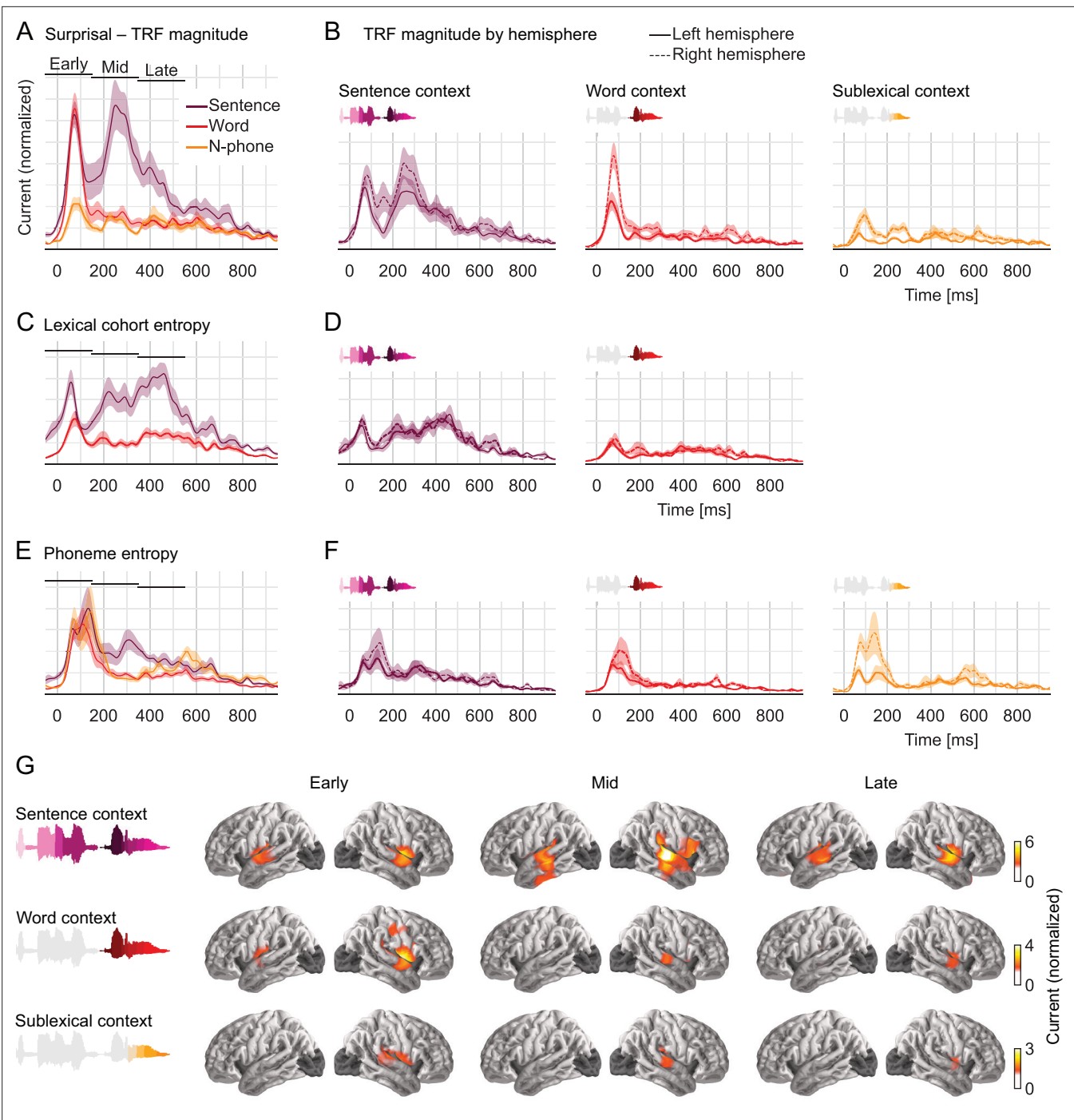

**Figure 5.** Early responses reflect parallel activation of all context models, later responses selectively reflect activity in the sentence-constrained model. (**A**) Current magnitude of temporal response functions (TRFs) to phoneme surprisal for each level of context (mean and within-subject standard error [*Loftus and Masson, 1994*]; *y*-axis scale identical in all panels of the figure). To allow fair comparison, all TRFs shown are from the same symmetric region of interest, including all current dipoles for which at least one of the three context models significantly improved the response predictions. Bars indicate time windows corresponding to source localizations shown in panel G. (**B**) When plotted separately for each hemisphere, relative lateralization of the TRFs is consistent with the lateralization of predictive power (*Figure 4*). (**C, D**) TRFs to lexical cohort entropy are dominated by the sentence context model. (**E, F**) TRFs to phoneme entropy are similar between context models, consistent with parallel use of different contexts in predictive models for upcoming speech. (**G**) All context models engage the superior temporal gyrus at early responses, midlatency responses incorporating the sentence context also engage more ventral temporal areas. Anatomical plots reflect total current magnitude associated with different levels of context representing early (−50 to 150 ms), midlatency (150–350 ms), and late (350–550 ms) responses. The color scale is adjusted for different predictors to avoid images dominated by the spatial dispersion characteristic of magnetoencephalography source estimates.

*Figure 5 continued on next page*

*Figure 5 continued*

The online version of this article includes the following source data for figure 5:

**Source data 1.** Temporal response function peak latencies in the early time window.

**Source data 2.** Pairwise tests of temporal response function time courses.

acoustic signal prior to the conventional phoneme onset (*Beddor et al., 2013*; *Salverda et al., 2003*). *Figure 5G* shows the anatomical distribution of responses related to the different levels of context.

Surprisal quantifies the incremental update to a context model due to new input. A brain response related to surprisal therefore indicates that the sensory input is brought to bear on a neural representation that uses the corresponding context model. Consequently, the latencies of brain responses related to surprisal at different context models are indicative of the underlying processing architecture. In an architecture in which information is sequentially passed to higher-level representations with broadening context models (*Figure 1B*), responses should form a temporal sequence from narrower to broader contexts. However, in contrast to this prediction, the observed responses to surprisal suggest that bottom-up information reaches representations that use the sentence- and word-level contexts *simultaneously*, at an early response peak (*Figure 5A*; peak in the early time window for sentence context: 78 ms, standard deviation (SD) = 24 ms; word context: 76 ms, SD = 11 ms). Sublexical surprisal is associated with a lower response magnitude overall, but also exhibits an early peak at 94 ms (SD = 26 ms). None of these three peak latencies differ significantly (all pairwise $t_{11} \leq 2.01$, $p \geq 0.065$). This suggests a parallel processing architecture in which different context representations are activated simultaneously by new input. Later in the timecourse the responses dissociate more strongly, with a large, extended response reflecting the sentence context, but not the word context starting at around 205 ms ($t_{max} = 5.27$, $p = 0.007$). The lateralization of the TRFs is consistent with the trend observed for predictive power: a symmetric response reflecting the unified sentence context, and more right-lateralized responses reflecting the more local contexts (*Figure 5B*).

The TRFs, convolved with the corresponding predictors generate partial predicted responses (*Figure 2C*). This reconstruction thus allows decomposing brain responses into component responses corresponding to different predictors. *Figure 6* uses this to simulate the responses corresponding to the different context models, illustrating several of the observations made above. As the sentence level

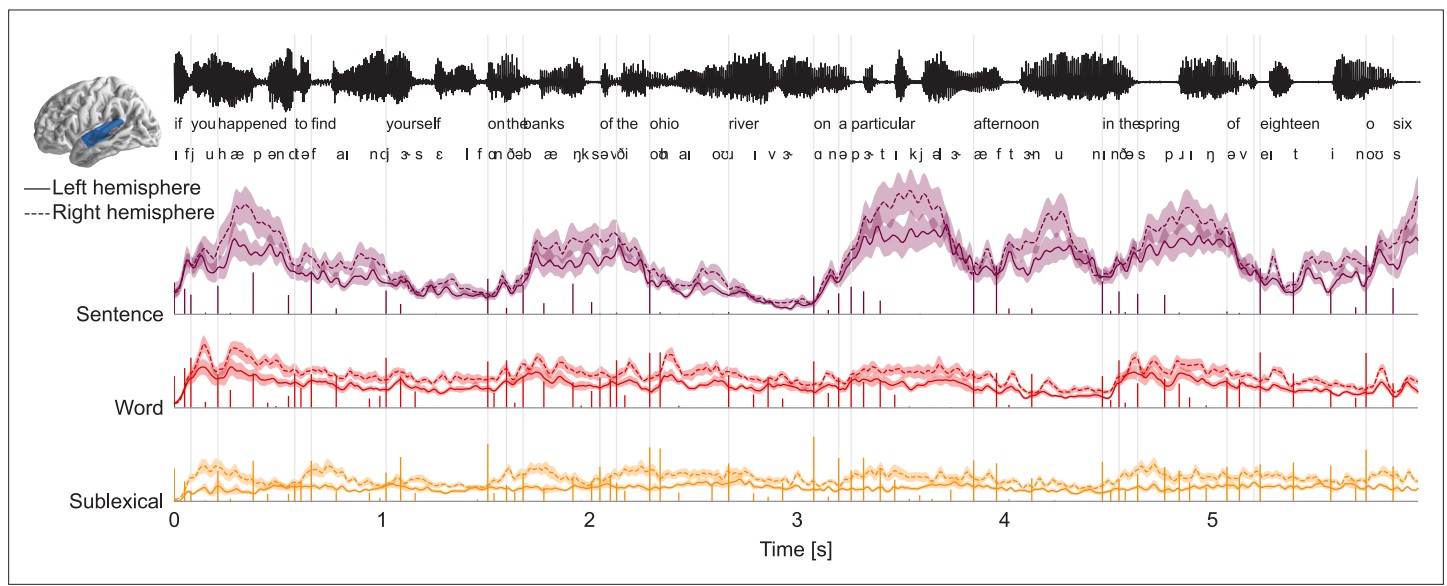

**Figure 6.** Decomposition of brain responses into context levels. Predicted brain responses related to processing different levels of context (combining surprisal and entropy predictors; mean and within-subject standard error). Stem plots correspond to surprisal for the given level. Slow fluctuations in brain responses are dominated by the sentence level, with responses occurring several hundred milliseconds after surprising phonemes, consistent with the high amplitude at late latencies in temporal response functions (TRFs). Partial predicted responses were generated for each context model by convolving the TRFs with the corresponding predictors; summing the predicted responses for the predictors corresponding to the same level; and extracting the magnitude (sum of absolute values) in the superior temporal gyrus region of interest.

has the most predictive power, so it also corresponds to higher amplitude responses than the other levels. Furthermore, the subsentence levels exhibit small modulations close to surprising phonemes, corresponding to the mainly brief, low latency TRFs. In contrast, the response corresponding to the sentence level is dominated by larger waves, lagging surprising phonemes by several hundred milliseconds, corresponding to the sustained, higher TRF amplitudes at later latencies.

## Sentence context dominates word recognition, all contexts drive phoneme predictions

Brain responses related to entropy indicate that neural processes are sensitive to uncertainty or competition in the interpretation of the speech input. Like surprisal, such a response suggests that the information has reached a representation that has incorporated the corresponding context. In addition, because entropy measures uncertainty regarding a categorization decision, the response to entropy can distinguish between different levels of categorization: uncertainty about the current word (cohort entropy) versus uncertainty about the next phoneme (phoneme entropy).

The TRFs to cohort entropy suggest a similar pattern as those to surprisal (*Figure 5C, D*). Both cohort representations are associated with an early peak (sentence context: 56 ms, SD = 28 ms; word context: 80 ms, SD = 23 ms), followed only in the sentence-constrained cohort by a later sustained effect. In contrast to surprisal, however, even early responses to cohort entropy are dominated by the sentence context ($t_{max}$ = 5.35, p = 0.004 at 43 ms; later responses: $t_{max}$ = 7.85, p < 0.001 at 461 ms). This suggests that lexical representations are overall most strongly activated in a model that incorporates the sentence context.

In contrast to surprisal and cohort entropy, the responses to phoneme entropy are similar for all levels of context, dominated by an early and somewhat broader peak (*Figure 5E, F*). There is still some indication of a second, later peak in the response to sentence-constrained phoneme entropy, but this might be due to the high correlation between cohort and phoneme entropy. A direct comparison of sentence-constrained cohort and phoneme entropy indicates that early processing is biased toward phoneme entropy (though not significantly) while later processing is biased toward cohort entropy ($t_{max}$ = 4.74, p = 0.017 at 231 ms).

In sum, the entropy results suggest that all context representations drive a predictive model for upcoming phonemes. This is reflected in a short-lived response in STG, consistent with the fast rate of phonetic information. Simultaneously, the incoming information is used to constrain the cohort of word candidates matching the current input, with lexical activations primarily driven by a unified model that incorporates the sentence context.

## Midlatency, sentence-constrained processing engages larger parts of the temporal lobe

Source localization suggests that early activity originates from the vicinity of the auditory cortex in the upper STG, regardless of context (*Figure 5G*). The precise source configuration in the right STG nevertheless differs between contexts in the early time window (sentence vs word: $F_{(175, 1925)}$ = 2.08, p < 0.001; word vs sublexical: $F_{(175, 1925)}$ = 5.99, p < 0.001). More notably, the sentence-based responses in the midlatency window recruits more sources, localized to the middle and inferior temporal lobe. Accordingly, the sentence-based responses in the midlatency window differs significantly from the early window (left hemisphere: $F_{(179, 1969)}$ = 1.72, p < 0.001; right hemisphere: $F_{(175, 1925)}$ = 5.48, p < 0.001). These results suggest that phonetic information initially engages a set of sources in the STG, while a secondary stage then engages more ventral sources that specifically represent the sentence context.

## No evidence for a trade-off between contexts

We interpret our results as evidence that different context models are maintained in parallel. An alternative possibility is that there is some trade-off between contexts used, and it only appears in the averaged data as if all models were operating simultaneously. This alternative predicts a negative correlation between the context models, reflecting the trade-off in their activation. No evidence was found for such a trade-off, as correlation between context models were generally neutral or positive across subjects and across time (*Figure 7*).

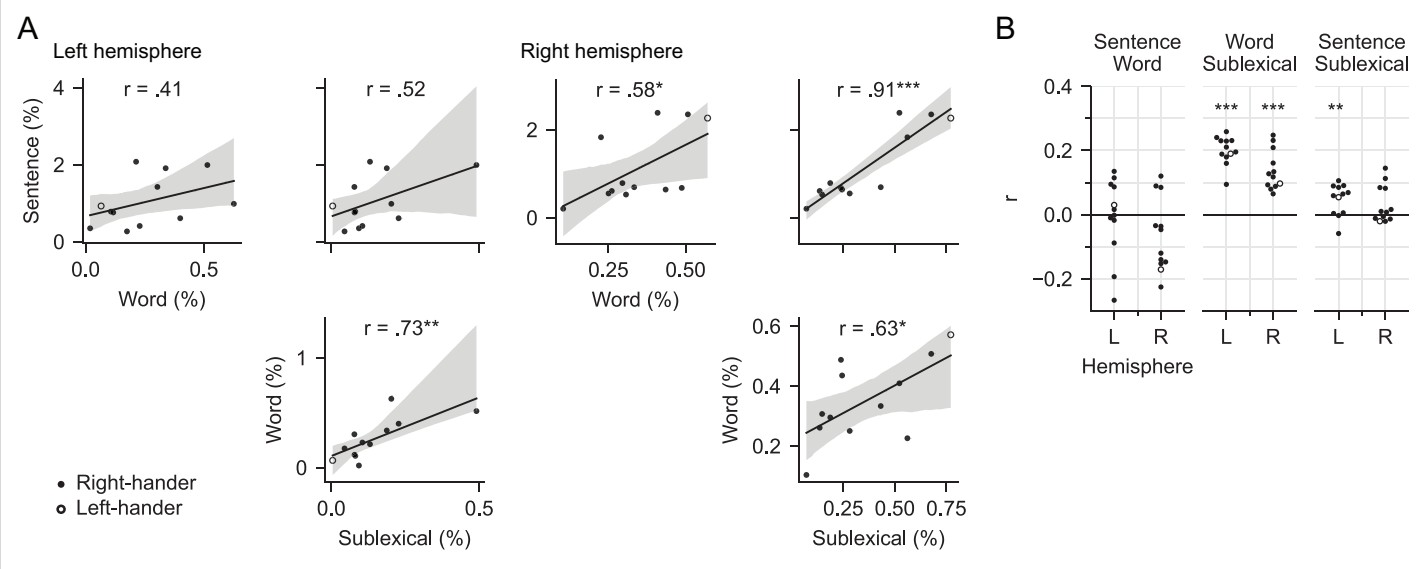

**Figure 7.** No evidence for a trade-off between context models. (**A**) Trade-off across subjects: testing the hypothesis that subjects differ in which context model they rely on. Each plot compares the predictive power of two context models in the mid/posterior superior temporal gyrus region of interest, each dot representing % explained for one subject. The line represents a linear regression with 95% bootstrap confidence interval (**Waskom, 2021**). None of the pairwise comparisons exhibits a negative correlation that would be evidence for a trade-off between reliance on different context models. Data from **Figure 4—source data 2**. (**B**) Trade-off over time: testing the hypothesis that subjects alternate over time in which context model they rely on. Each dot represents the partial correlation over time between the predictive power of two context models for one subject, controlling for predictive power of the full model. Correlations are shown separately for the left and the right hemisphere (L/R). Stars correspond to a one-sample *t*-tests of the null hypothesis that the average *r* across subjects is 0, that is, that the two context models are unrelated over time. None of the context models exhibited a significant negative correlation that would be evidence for a trade-off over time.

The online version of this article includes the following source data for figure 7:

**Source data 1.** Partial correlations over time for each subject (data for Panel B).

## Discussion

The present MEG data provide clear evidence for the existence of a neural representation of speech that is unified across representational hierarchies. This representation incrementally integrates phonetic input with information from the multiword context within about 100 ms. However, in addition to this globally unified representation, brain responses also show evidence of separate neural representations that use more local contexts to process the same input.

### Parallel representations of speech using different levels of context

The evidence for a unified global model suggests that there is a functional brain system that processes incoming phonemes while building a representation that incorporates constraints from the multiword context. A possible architecture for such a system is the unified global architecture shown in **Figure 1C**, in which a probabilistic representation of the lexical cohort mediates between sentence- and phoneme-level representations: the sentence context modifies the prior expectation for each word, which is in turn used to make low-level predictions about the phonetic input. While there are different possible implementations for such a system, the key feature is that the global sentence context is used to make predictions for and interpret low-level phonetic, possibly even acoustic (**Sohoglu and Davis, 2020**) input.

A second key result from this study, however, is evidence that this unified model is not the only representation of speech. Brain responses also exhibited evidence for two other, separate functional systems that process incoming phonemes while building representations that incorporate different, more constrained kinds of context: one based on a local word context, processing the current word with a prior based on context-independent lexical frequencies, and another based on the local phoneme sequence regardless of word boundaries. Each of these three functional systems generates its own predictions for upcoming phonemes, resulting in parallel responses to phoneme entropy. Each system

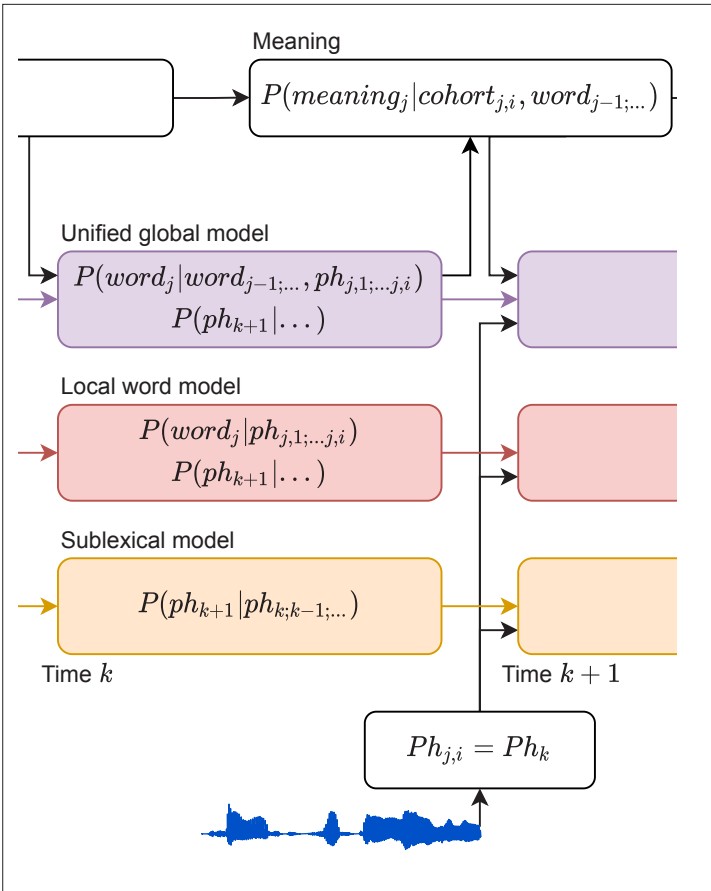

**Figure 8.** An architecture for speech perception with multiple parallel context models. A model of information flow, consistent with brain signals reported here. Brain responses associated with Information-theoretic variables provide separate evidence for each of the probability distributions in the colored boxes. From left to right, the three different context models (sentence, word, and sublexical) update incrementally as each phoneme arrives. The cost of these updates is reflected in the brain response related to surprisal. Representations also include probabilistic representations of words and upcoming phonemes, reflected in brain responses related to entropy.

is updated incrementally at the phoneme rate, reflected in early responses to surprisal. However, each system engages an at least partially different configuration of neural sources, as evidenced by the localization results.

Together, these results suggest that multiple predictive models process speech input in parallel. An architecture consistent with these observations is sketched in *Figure 8*: three different neural systems receive the speech input in parallel. Each representation is updated incrementally by arriving phonemes. However, the three systems differ in the extent and kind of context that they incorporate, each generating its own probabilistic beliefs about the current word and/or future phonemes. For instance, the sublexical model uses the local phoneme history to predict upcoming phonemes. The updates are incremental because the state of the model at time *k* + 1 is determined by the state of the model at time *k* and the phoneme input from time *k*. The same incremental update pattern applies to the sublexical, word, and sentence models.

A listener whose goal is comprehending a discourse-level message might be expected to rely primarily on the unified, sentence-constrained context model. Consistent with this, there is some evidence that this model has a privileged status. Among the linguistic models, the unified model has the most explanatory power and clearly bilateral representation (*Figure 4*). In addition, while activity in local models was short lived, the unified model was associated with extended activation for up to 600 ms and recruitment of more ventral regions of the temporal lobe (*Figure 5*). This suggests that the update in the unified model is normally more extensive than the local models, and could indicate that

the unified model most commonly drives semantic as well as form representations, while the short-lived local models might be restricted to form-based representations.

## Implications for speech processing

A longstanding puzzle in the comprehension literature has been why activation of candidates not supported by context is sometimes reported (*Swinney, 1979*; *Zwitserlood, 1989*), if top-down sentence context rapidly feeds down to early levels of speech perception. Parallel activation of lexical candidates based on sentence and word context models can explain these findings. Short-lived brain responses (up to 150 ms after phoneme onset) show evidence of parallel activation of sentence-constrained as well as sentence-independent word candidates. The coexistence of these two candidate sets can explain short-lived priming of sentence-inappropriate candidates. Whereas brain responses related to sentence-independent candidates are transient, brain responses related to sentence-appropriate candidates exhibit a secondary, sustained response (150–550 ms), explaining selective priming of sentence-appropriate candidates at longer delays.

If context-constrained candidates are immediately available, then why maintain alternative, sentence-independent candidates or even sublexical probabilistic phoneme sequences? One functional advantage might be faster recovery when sentence-based predictions turn out to be misleading. Such an effect has been described in reading, where contextually unexpected continuations are not associated with a proportional increase in processing cost (*Frisson et al., 2017*; *Luke and Christianson, 2016*).

Similarly, a representation of sublexical phoneme sequences might be functionally relevant when encountering input that is noisy or not yet part of the lexicon. Phoneme transition probabilities are generally higher within words than between words, such that low probability phoneme transitions are cues to word boundaries (*Cairns et al., 1997*; *Harris, 1955*). Statistical phoneme sequence models might thus play an important role in language acquisition by bootstrapping lexical segmentation of continuous speech (*Cairns et al., 1997*; *Chambers et al., 2003*; *Saffran et al., 1996*). Even in adult speech perception, they might have a similar function when encountering novel words, such as domain-specific vocabularies or personal names (*Norris and McQueen, 2008*). Finally, the linguistic context can be highly informative for phoneme recognition (*Hitczenko et al., 2020*), and different levels of context might make complementary contributions.

Our results suggest that the different context models operate in parallel, without an apparent trade-off, between subjects, or over time. However, our listening condition was also relatively uniform – listening to an audiobook in quiet background. It is conceivable that the different models play more differentiated roles in different listening conditions, for example in unpredictable conversations or speech in noise.

## Implications for word recognition

Perhaps the most surprising implication of our results is that multiple probabilistic cohort representations seem to be maintained in parallel. This is most directly implied by the result that two different cohort entropy predictors both explain unique variability in the data, one based on the lexical frequency of each candidate, and another based on the contextual likelihood of each candidate. This is inconsistent with models in which each lexical candidate is assigned a single 'activation' value (e.g., *Morton, 1969*). It might be more readily reconciled with models that distinguish between a lexical item's entry in long-term memory, and its instantiation as a token for parsing a speech signal, since such a mechanism allows for multiple tokens corresponding to the same lexical item (*McClelland and Elman, 1986*; *Norris, 1994*; *Norris and McQueen, 2008*). Yet, existing models generally are restricted to a single 'arena' for lexical competition, whereas our results imply the possibility that the competition plays out in parallel in at least partially different brain systems.

A second implication is that feedback from the sentence-level context can and does affect phoneme processing. The observed phoneme entropy effects indicate that phoneme-level expectations are modulated by the sentence context. This is inconsistent with some models of word recognition that assume a pure bottom-up process (e.g., *Norris et al., 2000*). However, at the same time, the parallel architecture we propose (*Figure 8*) addresses a central theoretical problem associated with feedback in multistage architectures: Bayesian accounts of perception suggest that listeners generate a prior, reflecting an estimate of future input, and compare this prior to the actual input to

compute a posterior probability, or interpretation of the sensory percept. In multistage architectures that allow different priors at sequential hierarchical levels (such as *Figure 1B*), higher levels receive the posterior interpretation of the input from the lower levels, rather than the unbiased input itself. This is suboptimal when considering a Bayesian model of perception, because the prior of lower-level systems is allowed to distort the bottom-up evidence before it is compared to the prior generated by higher levels (*Norris et al., 2016*). In contrast, the parallel representations favored by the evidence presented here allow undistorted bottom-up information to be directly compared with the context model for each definition of context. The parallel model can thus accommodate empirical evidence for feedback while avoiding this theoretical problem associated with sequential models.

## Evidence for graded linguistic predictions

There is broad agreement that language processing involves prediction, but the exact nature of these predictions is more controversial (*DeLong et al., 2005*; *Huettig, 2015*; *Nieuwland et al., 2020*; *Nieuwland et al., 2018*; *Pickering and Gambi, 2018*). Much of the debate is about whether humans can represent distributions over many likely items, or just predict specific items. Previous research showing an early influence of sentence context on speech processing has typically relied on specifically designed, highly constraining contexts which are highly predictive of a specific lexical item (*Holcomb and Neville, 2013*; *Connolly and Phillips, 1994*; *Van Petten et al., 1999*; *Rommers et al., 2013*). In such highly predictive contexts, listeners might indeed predict specific items, and such predictions might be linked to the left-lateralized speech productions system (*Federmeier, 2007*; *Pickering and Gambi, 2018*). However, such a mechanism would be less useful in more representative language samples, in which highly predictable words are rare (*Luke and Christianson, 2016*). In such situations of limited predictability, reading time data suggest that readers instead make graded predictions, over a large number of possible continuations (*Luke and Christianson, 2016*; *Smith and Levy, 2013*). Alternatively, it has also been suggested that what looks like graded predictions could actually be preactivation of specific higher-level semantic and syntactic features shared among the likely items (*Altmann and Kamide, 1999*; *Luke and Christianson, 2016*; *Matchin et al., 2019*; *Pickering and Gambi, 2018*; *Van Berkum et al., 2005*), without involving prediction of form-based representations. The present results, showing brain responses reflecting sentence-constrained cohort- and phoneme entropy, provide a new kind of evidence in favor of graded probabilistic and form-based predictions at least down to the phoneme level.

## Bilateral pathways to speech comprehension

Our results suggest that lexical/phonetic processing is largely bilateral. This is consistent with extensive clinical evidence for bilateral receptive language ability (*Gazzaniga and Sperry, 1967*; *Kutas et al., 1988*; *Poeppel, 2001*; *Hickok and Poeppel, 2007*), and suggestions that the right hemisphere might even play a distinct role in complex, real-world language processing (*Federmeier et al., 2008*; *Jung-Beeman, 2005*). In healthy participants, functional lateralization of sentence processing has been studied in reading using visual half-field presentation (*Federmeier and Kutas, 1999*). Overwhelmingly, results from these studies suggest that lexical processing in both hemispheres is dominated by sentence meaning (*Coulson et al., 2005*; *Federmeier et al., 2005*; *Federmeier and Kutas, 1999*; *Wlotko and Federmeier, 2007*). This is consistent with the strong bilateral representation of the unified model of speech found here. As in the visual studies, the similarity of the response latencies in the two hemispheres implies that right-hemispheric effects are unlikely to be due to interhemispheric transfer from the left hemisphere (*Figure 5*).

In addition to bilateral representation, however, our results also suggest that the two hemispheres differ with respect to the context models they entertain. Visual half-field reading studies have indicated a pattern of hemispheric differences, which has been interpreted as indicating that the left hemisphere processes language in a maximally context-sensitive manner, whereas the right hemisphere processes the sensory input in a bottom-up manner, unbiased by the linguistic context (*Federmeier, 2007*). Our results suggest a modification of this proposal, indicating that both hemispheres rely on sentence-based, graded predictions, but that the right hemisphere *additionally* maintains stronger representations of local contexts. Finally, lateralization might also depend on task characteristics such as stimulus familiarity (*Brodbeck et al., 2018a*), and in highly constraining contexts the left

hemisphere might engage the left-lateralized language production system to make specific predictions (*Federmeier, 2007*; *Pickering and Gambi, 2018*).

### Limitations of the sentence context model

We approximated the sentence context with a 5 gram model. This model provides an accurate estimate of the sum of local constraints, based on a context of the four preceding words only. However, it misses the more subtle influences of the larger context, both semantic constraints and syntactic long-range dependencies, which might make the sentence level even more different from the local context models. Furthermore, lexical *N* gram models conflate the influence of syntactic, semantic, and associative constraints (e.g., idioms). It is thus possible that work with more sophisticated language models can reveal an even more complex relationship between global and local contexts.

### Conclusions

Prior research on the use of context during language processing has often focused on binary distinctions, such as asking whether context is or is not used to predict future input. Such questions assumed a single serial or cascaded processing stream. Here, we show that this assumption might have been misleading, because different predictive models are maintained in parallel. Our results suggest that robust speech processing is based on probabilistic predictions using different context models in parallel, and cutting across hierarchical levels of representations.

## Materials and methods

### Participants

Twelve native speakers of English were recruited from the University of Maryland community (six female, six male, age mean = 21 years, range 19–23). None reported any neurological or hearing impairment. According to self-report using the Edinburgh Handedness Inventory (*Oldfield, 1971*), 11 were right handed and 1 left handed. All subjects provided written informed consent in accordance with the University of Maryland Institutional Review Board. Subjects either received course credit (*n* = 4) or were paid for their participation (*n* = 8). This sample size is comparable to the most directly relatable previous research which either had a similar number of subjects, *N* = 11 (*Donhauser and Baillet, 2020*) or a larger number of subjects but substantially less data per subject, *N* = 28 with single talker stimulus duration 8 min (*Brodbeck et al., 2018a*).

### Stimuli

Stimuli consisted of 11 excerpts from the audiobook version of *The Botany of Desire* by *Michael Pollan* (*Pollan, 2001*). Each excerpt was between 210 and 332 s long, for a total of 46 min and 44 s. Excerpts were selected to create a coherent narrative and were presented in chronological order to maximize deep processing for meaning.

### Procedure

During MEG data acquisition, participants lay in a supine position. They were allowed to keep their eyes open or closed to maximize subjective comfort and allow them to focus on the primary task of listening to the audiobook. Stimuli were delivered through foam pad earphones inserted into the ear canal at a comfortably loud listening level. After each segment, participants answered two to three questions relating to its content and had an opportunity to take a short break.

### Data acquisition and preprocessing

Brain responses were recorded with a 157 axial gradiometer whole head MEG system (KIT, Kanazawa, Japan) inside a magnetically shielded room (Vacuumschmelze GmbH & Co. KG, Hanau, Germany) at the University of Maryland, College Park. Sensors (15.5 mm diameter) are uniformly distributed inside a liquid-He dewar, spaced ~25 mm apart, and configured as first-order axial gradiometers with 50 mm separation and sensitivity better than 5 fT $Hz^{-1/2}$ in the white noise region (>1 kHz). Data were recorded with an online 200 Hz low-pass filter and a 60 Hz notch filter at a sampling rate of 1 kHz.

Recordings were preprocessed using mne-python (*Gramfort et al., 2014*). Flat channels were automatically detected and excluded. Extraneous artifacts were removed with temporal signal space

separation (*Taulu and Simola, 2006*). Data were filtered between 1 and 40 Hz with a zero-phase FIR filter (mne-python 0.20 default settings). Extended infomax independent component analysis (*Bell and Sejnowski, 1995*) was then used to remove ocular and cardiac artifacts. Responses time locked to the speech stimuli were extracted, low pass filtered at 20 Hz and resampled to 100 Hz.

Five marker coils attached to participants' head served to localize the head position with respect to the MEG sensors. Head position was measured at the beginning and at the end of the recording session and the two measurements were averaged. The FreeSurfer (*Fischl, 2012*) 'fsaverage' template brain was coregistered to each participant's digitized head shape (Polhemus 3SPACE FASTRAK) using rotation, translation, and uniform scaling. A source space was generated using fourfold icosahedral subdivision of the white matter surface, with source dipoles oriented perpendicularly to the cortical surface. Regularized minimum l2 norm current estimates (*Dale and Sereno, 1993*; *Hämäläinen and Ilmoniemi, 1994*) were computed for all data using an empty room noise covariance ($\lambda$ = 1/6). The TRF analysis was restricted to brain areas of interest by excluding the occipital lobe, insula, and midline structures based on the 'aparc' FreeSurfer parcellation (*Desikan et al., 2006*). Excluded areas are shaded gray in *Figure 4*. A preliminary analysis (see below) was restricted to the temporal lobe (superior, middle, and inferior temporal gyri, Heschl's gyrus, and superior temporal sulcus).

## Predictor variables
### Acoustic model
To control for brain responses to acoustic features, all models included an eight band gammatone spectrogram and an eight band acoustic onset spectrogram (*Brodbeck et al., 2020*), both covering frequencies from 20 to 5000 Hz in equivalent rectangular bandwidth space (*Heeris, 2018*) and scaled with exponent 0.6 (*Biesmans et al., 2017*).

### Word- and phoneme segmentation
A pronunciation dictionary was generated by combining the Carnegie-Mellon University pronunciation dictionary with the Montreal Forced Aligner (*McAuliffe et al., 2017*) dictionary and adding any additional words that occurred in the stimuli. Transcripts were then aligned to the acoustic stimuli using the Montreal Forced Aligner (*McAuliffe et al., 2017*) version 1.0.1. All models included control predictors for word onsets (equal value impulse at the onset of each word) and phoneme onsets (equal value impulse at the onset of each non word-initial phoneme).

### Context-based predictors
All experimental predictor variables consist of one value for each phoneme and were represented as a sequence of impulses at all phoneme onsets. The specific values were derived from three different linguistic context models.

### Sublexical context model
The complete SUBTLEX-US corpus (*Brysbaert and New, 2009*) was transcribed by substituting the pronunciation for each word and concatenating those pronunciations across word boundaries (i.e., no silence between words). Each line was kept separate since lines are unordered in the SUBTLEX corpus. The resulting phoneme sequences were then used to train a 5 gram model using KenLM (*Heafield, 2011*). This 5 gram model was then used to derive phoneme surprisal and entropy.

The surprisal of experiencing phoneme $ph_k$ at time point $k$ is inversely related to the likelihood of that phoneme, conditional on the context (measured in bits): $I\left(ph_k\right) = -log_2\left(p\left(ph_k|context\right)\right)$. In the case of the 5 phone model this context consists of the preceding four phonemes, $ph_{k-4;...k-1}$.

The entropy $H$ (Greek Eta) at phoneme position $ph_k$ reflects the uncertainty of what the next phoneme, $ph_{k+1}$ will be. It is defined as the expected (average) surprisal at the next phoneme, $H_{ph}\left(ph_k\right) = -\sum_{ph}^{phonemes} p\left(ph_{k+1} = ph|context\right) \log_2\left(p\left(ph_{k+1} = ph|context\right)\right)$. Based on the 5 phone model, the context here is $ph_{k-3;...k}$.

### Word context model
The word context model takes into account information from all phonemes that are in the same word as, and precede the current phoneme (*Brodbeck et al., 2018a*) and is based on the cohort model of word perception (*Marslen-Wilson, 1987*). At word onset, the prior for each word is proportional

to its frequency in the Corpus of Contemporary American English (COCA; **Davies, 2015**). With each subsequent phoneme, the probability for words that are inconsistent with that phoneme is set to 0, and the remaining distribution is renormalized. Phoneme surprisal and entropy are then calculated as above, but with the context being all phonemes in the current word so far. In addition, lexical entropy is calculated at each phoneme position as the entropy in the distribution of the cohort $H_w\left(ph_{j,i}\right) = -\sum_{word}^{lexicon} p\left(word_j = word|context\right)\log_2\left(p\left(word_j = word|context\right)\right)$ where $j$ is the index of the word, $i$ is the index of the current phoneme within word $j$, and the context consists of phonemes $ph_{j,1;...,j,i-1}$.

This context model thus allows two different levels of representation, phonemes and words, and two corresponding entropy values, phoneme entropy and lexical entropy. Yet, we only include one version of surprisal. The reason for this is that calculating surprisal over phonemes or over words leads to identical results. This is because the $k$th phoneme of a word, together with the cohort at phoneme $k-1$ exhaustively defines the cohort at phoneme $k$: $p\left(ph_k|cohort_{k-1}\right) \equiv p\left(cohort_k|cohort_{k-1}\right)$.

### Sentence context model

The sentence context model was implemented like the lexical context model, but with the addition of lexical priors based on the 5 gram word context. A 5 gram model was trained on COCA (**Davies, 2015**) with KenLM (**Heafield, 2011**). Then, at the onset of each word, the cohort was initialized with each word's prior set to its probability given the four preceding words in the 5 gram model.

## Deconvolution

Deconvolution and statistical analysis were performed with Eelbrain (**Brodbeck et al., 2021**) and additional scripts available at https://github.com/christianbrodbeck/TRF-Tools (**Brodbeck, 2021**).

mTRFs were computed independently for each subject and each virtual current source (**Brodbeck et al., 2018b**; **Lalor et al., 2009**). The neural response at time $t$, $y_t$ was predicted jointly from $N$ predictor time series $x_{i,t}$ convolved with a corresponding mTRF $h_{i_t}$ of length $T$:

$$\hat{y}_t = \sum_i^N \sum_\tau^T h_{n,\tau} \cdot x_{i,t-\tau}$$

mTRFs were generated from a basis of 50 ms wide Hamming windows centered at $T = \left[-100, \ldots, 1000\right)$ ms. For estimating mTRFs, all responses and predictors were standardized by centering and dividing by the mean absolute value.

For estimation using fourfold cross-validation, each subject's data were concatenated along the time axis and split into four contiguous segments of equal length. The mTRFs for predicting the responses in each segment were trained using coordinate descent (**David et al., 2007**) to minimize the l1 error in the three remaining segments. For each test segment there were three training runs, with each of the remaining segments serving as the validation segment once. In each training run, the mTRF was iteratively modified based on the maximum error reduction in two training segments (the steepest coordinate descent) and validated based on the error in the validation segment. Whenever a training step caused an increase of error in the validation segment, the TRF for the predictor responsible for the increase was frozen. Training continued until the whole mTRF was frozen. The three mTRFs from the three training runs were then averaged to predict responses in the left-out testing segment.

## Model comparisons

Model quality was quantified through the l1 norm of the residuals. For this purpose, the predicted responses for the four test segments, each based on mTRFs estimated on the other three segments, were concatenated again. To compare the predictive power of two models, the difference in the residuals of the two models was calculated at each virtual source dipole. This difference map was smoothed (Gaussian window, SD = 5 mm) and tested for significance using a mass-univariate one-sample $t$-test with threshold-free cluster enhancement (TFCE) (**Smith and Nichols, 2009**) and a null distribution based on the full set of 4095 possible permutations of the 12 difference maps. For effect size comparison we report $t_{max}$, the largest $t$-value in the significant (p ≤ 0.05) area.

The full model consisted of the following predictors: acoustic spectrogram (eight bands); acoustic onset spectrogram (eight bands); word onsets; phoneme onsets; sublexical context model (phoneme

surprisal and phoneme entropy); lexical context model (phoneme surprisal, phoneme entropy, and word entropy); sentence context model (phoneme surprisal, phoneme entropy, and word entropy).

For each of the tests reported in *Figure 4*, mTRFs were reestimated using a corresponding subset of the predictors in the full model. For instance, to calculate the predictive power for a given level of context, the model was refit using all predictors except the predictors of the level under investigation. Each plot thus reflects the variability that can *only* be explained by the level in question. This is generally a conservative estimate for the predictive power because it discounts any explanatory power based on variability that is shared with other predictors. In order to determine the predictive power of linguistic processing in general, we also fit a model excluding all eight information-theoretic predictors from the three levels combined.

To express model fits in a meaningful unit, the explainable variability was estimated through the largest possible explanatory power of the full model (maximum across the brain of the measured response minus residuals, averaged across subjects). All model fits were then expressed as % of this value. For visualization, brain maps are not masked by significance to accurately portray the continuous nature of MEG source estimates.

## Tests of lateralization

For spatiotemporal tests of lateralization (*Figure 4A, D*), the difference map was first morphed to the symmetric 'fsaverage_sym' brain (*Greve et al., 2013*), and the data from the right hemisphere were morphed to the left hemisphere. Once in this common space, a mass-univariate repeated measures *t*-test with TFCE was used to compare the difference map from the left and right hemisphere.

## Region of interest analysis

To allow for univariate analyses of predictive power, a ROI was used including a region responsive to all context models (white outline in *Figure 4A*). This ROI was defined as the posterior 2/3 of the combined Heschl's gyrus and STG 'aparc' label, separately in each hemisphere.

To compare relative lateralization in this ROI (*Figure 4E*), the predictive power in each hemisphere's ROI was rectified (values smaller than 0 were set to 0). The lateralization index (LI) was then computed as $LI = R / (L + R)$.

## Tests of localization difference

A direct comparison of two localization maps can have misleading results due to cancellation between different current sources (*Lütkenhöner, 2003*) as well as the spatially continuous nature of MEG source estimates (*Bourguignon et al., 2018*). However, a test of localization difference is possible due to the additive nature of current sources (*McCarthy and Wood, 1985*). Specifically, for a linear inverse solver as used here, if the relative amplitude of a configuration of current sources is held constant, the topography of the resulting source localization is also unchanged. Consequently, we employed a test of localization difference that has the null hypothesis that the topography of two effect in source space is the same (*McCarthy and Wood, 1985*). Localization tests were generally restricted to an area encompassing the major activation seen in *Figure 4*, based on 'aparc' labels (*Desikan et al., 2006*): the posterior 2/3 of the superior temporal gyrus and Heschl's gyrus combined, the superior temporal sulcus, and the middle 3/5 of the middle temporal gyrus. For each map, the values in this area were extracted and *z*-scored (separately for each hemisphere). For each comparison, the two *z*-scored maps were subtracted, and the resulting difference map was analyzed with a one-way repeated measures ANOVA with factor source location (left hemisphere: 180 sources; right hemisphere: 176 sources). According to the null hypothesis, the two maps should be (statistically) equal, and the difference map should only contain noise. In contrast, a significant effect of source location would indicate that the difference map reflects a difference in topography that is systematic between subjects.

## TRF analysis

For the analysis of the TRFs, all 12 mTRFs estimated for each subject were averaged (four test segments × three training runs). TRFs were analyzed in the normalized scale that was used for model estimation.

### TRF time course

To extract the time course of response functions, an ROI was generated including all virtual current sources for which at least one of the three context models significantly improved the response predictions. To allow a fair comparison between hemispheres, the ROI was made symmetric by morphing it to the 'fsaverage_sym' brain (*Greve et al., 2013*) and taking the union of the two hemispheres. With this ROI, the magnitude of the TRFs at each time point was then extracted as the sum of the absolute current values across source dipoles. These time courses were resampled from 100 Hz, used for the deconvolution, to 1000 Hz for visualization and for more accurate peak time extraction. Peak times were determined by finding the maximum value within the early time window (−50 to 150 ms) for each subject. Time courses were statistically compared using mass-univariate related measures *t*-tests, with a null distribution based on the maximum statistic in the 4095 permutations (no cluster enhancement).

### TRF localization

To analyze TRF localization, TRF magnitude was quantified as the summed absolute current values in three time windows, representing early (−50 to 150 ms), midlatency (150 to 350 ms), and late (350–550 ms) responses (see *Figure 5*). Maps were smoothed (Gaussian window, SD = 5 mm) and tested for localization differences with the same procedure as described above (tests of localization difference).

## Analysis of trade-off between context models

Several analyses were performed to detect a trade-off between the use of the different context models.

### Trade-off by subject

One possible trade-off is between subjects: some subjects might rely on sentence context more than local models, whereas other subjects might rely more on local models. For example, for lexical processing, this hypothesis would predict that for a subject for whom the sentence context model is more predictive, the lexical context model should be less and vice versa. According to this hypothesis, the predictive power of the different context models should be negatively correlated across subjects. To evaluate this, we computed correlations between the predictive power of the different models in the mid/posterior STG ROI (see *Figure 7A* ).

### Trade-off over time

A second possible trade-off is across time: subjects might change their response characteristics over time to change the extent to which they rely on the lower- or higher-level context. For example, the depth of processing of meaningful speech might fluctuate with the mental state of alertness. According to this hypothesis, the predictive power of the different context models should be anticorrelated over time. To evaluate this, we calculated the residuals for the different model fits for each time point, $res_t = abs\left(y_t - \hat{y}_t\right)$, aggregating by taking the mean in the mid/posterior STG ROI (separately or each subject). The predictive power was calculated for each model by subtracting the residuals of the model from the absolute values of the measured data (i.e., the residuals of a null model without any predictor). The predictive power for each level of context was then computed by subtracting the predictive power of a corresponding reduced model, lacking the given level of context, from the predictive power of the full model. Finally, to reduce the number of data points the predictive power was summed in 1 s bins.

For each subject, the trade-off between each pair of contexts was quantified as the partial correlation (*Vallat, 2018*) between the predictive power of the two contexts, controlling for the predictive power of the full model (to control for MEG signal quality fluctuations over time). To test for a significant trade-off, a one-sample *t*-test was used for the correlation between each pair of contexts in each hemisphere, with the null hypothesis that the correlation over time is 0 (see *Figure 7B* ).

## Additional information

### Funding

| Funder | Grant reference number | Author |
| --- | --- | --- |
| University of Maryland | BBI Seed Grant | Jonathan Z Simon<br>Ellen Lau<br>Christian Brodbeck<br>Aura AL Cruz Heredia |
| National Science Foundation | BCS-1749407 | Ellen Lau |
| National Institutes of Health | R01DC014085 | Jonathan Z Simon<br>Christian Brodbeck |
| National Science Foundation | SMA-1734892 | Jonathan Z Simon |
| Office of Naval Research | MURI Award N00014-18-1-2670 | Philip Resnik<br>Shohini Bhattasali |
| National Science Foundation | BCS-1754284 | Christian Brodbeck |

The funders had no role in study design, data collection, and interpretation, or the decision to submit the work for publication.

### Author contributions
Christian Brodbeck, Conceptualization, Data curation, Formal analysis, Investigation, Methodology, Project administration, Resources, Software, Validation, Visualization, Writing – original draft, Writing – review and editing; Shohini Bhattasali, Conceptualization, Data curation, Methodology, Resources, Writing – review and editing; Aura AL Cruz Heredia, Conceptualization, Data curation, Investigation; Philip Resnik, Conceptualization, Methodology, Supervision, Writing – review and editing; Jonathan Z Simon, Conceptualization, Funding acquisition, Methodology, Project administration, Resources, Supervision, Writing – review and editing; Ellen Lau, Conceptualization, Funding acquisition, Methodology, Project administration, Supervision, Writing – review and editing

### Author ORCIDs
Christian Brodbeck ⬤ http://orcid.org/0000-0001-8380-639X
Shohini Bhattasali ⬤ http://orcid.org/0000-0002-6767-6529
Philip Resnik ⬤ http://orcid.org/0000-0002-6130-8602
Jonathan Z Simon ⬤ http://orcid.org/0000-0003-0858-0698

### Ethics
The study was approved by the IRB of the University of Maryland under the protocol titled 'MEG Studies of Speech and Language Processing' (reference # 01153), on August 22, 2018 and September 9, 2019 (approval duration: 1 year). All participants provided written informed consent prior to the start of the experiment.

### Decision letter and Author response
Decision letter https://doi.org/10.7554/eLife.72056.sa1
Author response https://doi.org/10.7554/eLife.72056.sa2

## Additional files

### Supplementary files
• Transparent reporting form

### Data availability
The raw data and predictors used in this study are available for download from Dryad at https://doi.org/10.5061/dryad.nvx0k6dv0.

The following dataset was generated:

| Author(s) | Year | Dataset title | Dataset URL | Database and Identifier |
|---|---|---|---|---|
| Brodbeck C, Bhattasali S, Cruz Heredia A, Resnik P, Simon J, Lau E | 2022 | Data from: Parallel processing in speech perception with local and global representations of linguistic context | http://dx.doi.org/10.5061/dryad.nvx0k6dv0 | Dryad Digital Repository, 10.5061/dryad.nvx0k6dv0 |

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
