## [Editor Report]

To comprehend speech efficiently, the brain predicts what comes next as sentences unfold. In this study, Brodbeck and colleagues asked at which scale predictive processing helps the analysis of speech. The authors combined magnetoencephalography with state-of-the-art analyses (multivariate Temporal Response Functions) and information-theoretic measures (entropy, surprisal) to test distinct contextual speech models at three hierarchical processing levels. The authors report evidence for the coexistence of hierarchical and parallel speech processing supporting the independent contribution of local (e.g. sublexical) and global (e.g. sentences) contextual probabilities to the analysis of speech.

---

## [Decision Letter]

**Decision letter after peer review:**

Thank you for submitting your article "Parallel processing in speech perception: Local and global representations of linguistic context" for consideration by *eLife*. Your article has been reviewed by 3 peer reviewers, one of whom is a member of our Board of Reviewing Editors, and the evaluation has been overseen by Barbara Shinn-Cunningham as the Senior Editor. The following individual involved in review of your submission has agreed to reveal their identity: Jonathan Brennan (Reviewer #2).

Essential revisions:

The sample size (N=12) appears like a low number, and authors should rationalize their sample choice with a power analysis, eventually illustrate with single-participant level data or explain why, in light of the paradigmatic strategy and analyses performed, this sample size is reasonable.

Reviewer 2 suggested that yuo contextualize a bit better how outcomes of Figure 7 fits other models (e.g. TRACE vs. RACE) and how the authors' novel observations update or modify existing architectures in the field.

Reviewer 3 questioned in his first main comment the choice of number of terms in the models being tested. The authors may wish to carefully address possible shortcomings of how more global models may leave room for local models to capture variance.

Reviewer 1 would like to see some justifications about why the analysis of phase-locked activity vs. induced responses is informative, and whether the latter could reveal additional insights if at all.

Additional suggestions made by all Reviewers should help clarify and streamline the manuscript further. Please keep in mind that the audience in *eLife* is diverse and readers may not necessarily be expert in (neuro)linguistics or technically versed with MEG. The overall flow of the manuscript can be streamlined a bit so as to clarify the complexity of some analyses that Reviewers 2 and 3 pointed out (some snippets are provided by Reviewer 1).

*Reviewer #2 (Recommendations for the authors):*

I had a few comments that I hope might help to make this paper even more impactful.

First, Figure 7 offers a boxology of the parallel processing architecture the authors believe is consistent with the data. Overall I'm pretty sympathetic to this view, but I would have liked to see the Discussion section better connect these conclusions with the existing literature. As presented, the reader might take the Figure 7 architecture to be a totally new model. I think it would be more appropriate to see how this updates or refines existing models. Specifically, I found myself reading the Discussion section through the lens of the late 90s debate on lexical access, specifically the TRACE model with fully interactive access, as compared to the RACE model of fully bottom-up access. I think the existing model can be recast as an extension of TRACE, but perhaps with the addition of "outputs" at each intermediate level (not just at the top?) I may not be exactly right here, but the upshot is I'd appreciate some extra handholding here for the reader to see how this architecture updates existing theories.

Second, the right lateralization of lower level effects seems to warrant further discussion. The interpretatin of these seems to emphasize the bilateral nature of speech perception – no arguments there – but the data actually favor a right-hemisphere bias which is unexpected to me (cf. the Giraud and Poeppel model for speech perception placed phoneme-level analysis predominantly in the left hemisphere).

Third, at N=12 the sample size is relatively low for 2021, and some key statistics are only reported as t_max. Together, I'm a little concerned that this may be a bit anti-conservative. At the least, I would like to see the statistics for reliable effects reproted as ranges (t_min – t_max). Increasing the N of the study would be great, but I understand if it is not feasible.

Figure 1: Where does meaning(j,i) come from? The red coloring seems to indicate it is the output of the sentence-level box, but that isn't clear to me from the sentence(i,j) notation.

ln. 255-256 – "While surprisal depends on the conditional probability of a discrete event and is agnostic to the underlying unit of representation". I don't understand this point. Both surprisal and entropy are calculated over distributions of some particular representation (P(phoneme_i|phoneme_i-1) ! = P(phoneme_i|word_j)… P(phoneme|…) ! = P(word|…)) I'm afraid I'm missing the intended point.

ln 702-704: I'm having trouble understanding the test for localization differences. I gather that the analysis takes source amplitude differences (180 or 176) per participant, and subjected these to a one-way anova, which was repeated for each pair of conditions. If so, shouldn't the DF for the F-test be (179, 11) or (175,11)? Instead, ln. 294-295 gives F(175,1925) and F(179 , 1969). I don't understand where that residual DF is coming from.

---

## [Author Response]

Essential revisions:The sample size (N=12) appears like a low number, and authors should rationalize their sample choice with a power analysis, eventually illustrate with single-participant level data or explain why, in light of the paradigmatic strategy and analyses performed, this sample size is reasonable.

We appreciate and share the reviewers’ concern with statistical power and have made several modifications to better explain and rationalize our choices.

First, to contextualize our study: The sample size is similar to the most comparable published study, which had 11 participants (Donhauser
and
Baillet,
2020). Our own previous study (Brodbeck
et
al.,
2018) had more participants (28) but only a fraction of the data per subject (8 minutes of speech in quiet, vs. 47 minutes in the present dataset). We added this consideration to the Methods/Participants section.

We also added a table with effect-sizes for all the main predictors to make that information more accessible (Table 1). This suggests that the most relevant effects have Cohen’s *d* > 1. With our sample size 12, we had 94% power to detect an effect with *d* = 1, and 99% power to detect an effect with *d* = 1.2. This post-hoc analysis suggests that our sample was adequately powered for the intended purpose.

Finally, all crucial model comparisons are accompanied by swarm-plots that show each subject as a separate dot, thus showing that these comparisons are highly reproducible across participants (note that there rarely are participants with model difference below 0, indicating that the effects are all seen in most subjects).

Reviewer 2 suggested that yuo contextualize a bit better how outcomes of Figure 7 fits other models (e.g. TRACE vs. RACE) and how the authors' novel observations update or modify existing architectures in the field.

Please see the corresponding section for Reviewer #2 (Recommendations for the authors).

Reviewer 3 questioned in his first main comment the choice of number of terms in the models being tested. The authors may wish to carefully address possible shortcomings of how more global models may leave room for local models to capture variance.

Please see the corresponding section for Reviewer #3 (Public Review).

Reviewer 1 would like to see some justifications about why the analysis of phase-locked activity vs. induced responses is informative, and whether the latter could reveal additional insights if at all.

Please see the corresponding section for Reviewer #1 (Public Review).

Additional suggestions made by all Reviewers should help clarify and streamline the manuscript further. Please keep in mind that the audience in eLife is diverse and readers may not necessarily be expert in (neuro)linguistics or technically versed with MEG. The overall flow of the manuscript can be streamlined a bit so as to clarify the complexity of some analyses that Reviewers 2 and 3 pointed out (some snippets are provided by Reviewer 1).

We thank the reviewers for many great suggestions to make the manuscript more accessible. We have revised the manuscript to reduce reliance on terminology, and incorporated all the other suggestions (see the responses to the many individual suggestions below).

Reviewer #2 (Recommendations for the authors):I had a few comments that I hope might help to make this paper even more impactful.First, Figure 7 offers a boxology of the parallel processing architecture the authors believe is consistent with the data. Overall I'm pretty sympathetic to this view, but I would have liked to see the Discussion section better connect these conclusions with the existing literature. As presented, the reader might take the Figure 7 architecture to be a totally new model. I think it would be more appropriate to see how this updates or refines existing models. Specifically, I found myself reading the Discussion section through the lens of the late 90s debate on lexical access, specifically the TRACE model with fully interactive access, as compared to the RACE model of fully bottom-up access. I think the existing model can be recast as an extension of TRACE, but perhaps with the addition of "outputs" at each intermediate level (not just at the top?) I may not be exactly right here, but the upshot is I'd appreciate some extra handholding here for the reader to see how this architecture updates existing theories.

We have added a *Discussion section* called *Implications for word recognition* to discuss implications for existing models more explicitly. We are reluctant to draw stronger comparisons with computational models such as TRACE, because such models might have emergent properties that are not straightforward to deduce from their architecture, and instead require careful analysis of model behavior (e.g.
Luthra
et
al.,
2021). However, we share the reviewer’s interest and such work is underway (e.g. Brodbeck
et al., 2021)**.**

Second, the right lateralization of lower level effects seems to warrant further discussion. The interpretation of these seems to emphasize the bilateral nature of speech perception – no arguments there – but the data actually favor a right-hemisphere bias which is unexpected to me (cf. the Giraud and Poeppel model for speech perception placed phoneme-level analysis predominantly in the left hemisphere).

You are right to point this out. In order to verify this observation we performed an additional model test of linguistic processing in general (test the predictive power of the full model compared with a model excluding all linguistic predictors). It turns out that overall, linguistic processing is not significantly lateralized. This might seem counterintuitive given the significant lateralization of two out of three context models. However, it is important that the tests of the individual models partial out variability that can *only* be explained by the respective model, and so the test for the combined linguistic model thus likely explains more than the sum of the three individual comparisons, because it also includes variability that is shared between two or more of the individual models. We have added this test to the relevant Results section (Different context models affect different neural processes).

An additional consideration is that our method for estimating speech tracking is disproportionately sensitive to slow cortical frequencies below 10 Hz (Ding
et
al.,
2014) and such low frequencies might be inherently stronger in the right hemisphere (Giraud
et
al.,
2007). In our interpretation we thus emphasize relative patterns of lateralization, and are cautious about interpreting the absolute lateralization. Most importantly, our results suggest that speech perception is bilateral, but with different properties in each hemisphere, in a way that is consistent with findings based on a different methodology (as discussed under *Bilateral pathways to speech comprehension*).

Third, at N=12 the sample size is relatively low for 2021, and some key statistics are only reported as t_max. Together, I'm a little concerned that this may be a bit anti-conservative. At the least, I would like to see the statistics for reliable effects reported as ranges (t_min – t_max). Increasing the N of the study would be great, but I understand if it is not feasible.

Please see response in Essential revisions.

Figure 1: Where does meaning(j,i) come from? The red coloring seems to indicate it is the output of the sentence-level box, but that isn't clear to me from the sentence(i,j) notation.

Thank you for noticing this inconsistency, it should have said sentence(j-1) to invoke the last state of the higher level.

ln. 255-256 – "While surprisal depends on the conditional probability of a discrete event and is agnostic to the underlying unit of representation". I don't understand this point. Both surprisal and entropy are calculated over distributions of some particular representation (P(phoneme_i|phoneme_i-1) ! = P(phoneme_i|word_j)… P(phoneme|…) ! = P(word|…)) I'm afraid I'm missing the intended point.

This is indeed a tricky point to explain without equations, and we did not do it justice. We have made the relevant section more explicit, and we have also added the details with equations to the Methods section (*Lexical context model* subsection) and point the reader to this section from the main text. Based on a suggestion from Reviewer 3 we have also added the formal definitions to the Introduction, which further clarifies the distinction between phoneme and cohort entropy.

ln 702-704: I'm having trouble understanding the test for localization differences. I gather that the analysis takes source amplitude differences (180 or 176) per participant, and subjected these to a one-way anova, which was repeated for each pair of conditions. If so, shouldn't the DF for the F-test be (179, 11) or (175,11)? Instead, ln. 294-295 gives F(175,1925) and F(179 , 1969). I don't understand where that residual DF is coming from.

Please note that these are within-subject ANOVAs, so the denominator df is (n_subjects – 1) • (n_treatments – 1) = 11 • 175 = 1925 and 11 • 179 = 1969, respectively (e.g. Rutherford, 2001, p. 71).

References

Brodbeck C, Gaston P, Luthra S, Magnuson JS. 2021. Discovering computational principles in models and brains.

Brodbeck C, Hong LE, Simon JZ. 2018. Rapid Transformation from Auditory to Linguistic Representations of Continuous Speech. Curr Biol 28:3976-3983.e5.

doi:10.1016/j.cub.2018.10.042

Ding N, Chatterjee M, Simon JZ. 2014. Robust cortical entrainment to the speech envelope relies on the spectro-temporal fine structure. NeuroImage 88:41–46.

doi:10.1016/j.neuroimage.2013.10.054

Donhauser PW, Baillet S. 2020. Two Distinct Neural Timescales for Predictive Speech Processing. Neuron 105:385-393.e9. doi:10.1016/j.neuron.2019.10.019

Giraud A-L, Kleinschmidt A, Poeppel D, Lund TE, Frackowiak RSJ, Laufs H. 2007. Endogenous Cortical Rhythms Determine Cerebral Specialization for Speech Perception and Production. Neuron 56:1127–1134. doi:10.1016/j.neuron.2007.09.038

Gulordava K, Bojanowski P, Grave E, Linzen T, Baroni M. 2018. Colorless Green Recurrent Networks Dream HierarchicallyProceedings of the 2018 Conference of the North American Chapter of the Association for Computational Linguistics: Human Language Technologies, Volume 1 (Long Papers). Presented at the NAACL-HLT 2018. New Orleans, Louisiana: Association for Computational Linguistics. pp. 1195–1205. doi:10.18653/v1/N18-1108

Luthra S, Li MYC, You H, Brodbeck C, Magnuson JS. 2021. Does signal reduction imply predictive coding in models of spoken word recognition? Psychon Bull Rev.

doi:10.3758/s13423-021-01924-x

Rutherford A. 2001. Introducing ANOVA and ANCOVA: a GLM approach, Introducing statistical methods. London ; Thousand Oaks, Calif.: SAGE.

Shain C, Blank IA, van Schijndel M, Schuler W, Fedorenko E. 2020. fMRI reveals language-specific predictive coding during naturalistic sentence comprehension. Neuropsychologia 138:107307. doi:10.1016/j.neuropsychologia.2019.107307

Willems RM, Van der Haegen L, Fisher SE, Francks C. 2014. On the other hand: including left-handers in cognitive neuroscience and neurogenetics. Nat Rev Neurosci 15:193–201. doi:10.1038/nrn3679